# Endosomal Rab cycles regulate Parkin-mediated mitophagy

Koji Yamano[1,2]*, Chunxin Wang[2], Shireen A Sarraf[2], Christian Münch[3,4], Reika Kikuchi[1], Nobuo N Noda[5], Yohei Hizukuri[6], Masato T Kanemaki[7,8,9], Wade Harper[3], Keiji Tanaka[10], Noriyuki Matsuda[1]*, Richard J Youle[2]*

[1]Ubiquitin Project, Tokyo Metropolitan Institute of Medical Science, Tokyo, Japan; [2]Biochemistry Section, Surgical Neurology Branch, National Institute of Neurological Disorders and Stroke, National Institutes of Health, Bethesda, United States; [3]Department of Cell Biology, Harvard Medical School, Boston, United States; [4]Institute of Biochemistry II, School of Medicine, Goethe University, Frankfurt, Germany; [5]Institute of Microbial Chemistry, Tokyo, Japan; [6]Institute for Frontier Life and Medical Sciences, Kyoto University, Kyoto, Japan; [7]Division of Molecular Cell Engineering, National Institute of Genetics, Research Organization of Information and Systems, Mishima, Japan; [8]Department of Genetics, SOKENDAI, Mishima, Japan; [9]Division of Molecular Cell Engineering, National Institute of Genetics, ROIS, Mishima, Japan; [10]Laboratory of Protein Metabolism, Tokyo Metropolitan Institute of Medical Science, Tokyo, Japan

*For correspondence:
yamano-kj@igakuken.or.jp (KY);
matsuda-nr@igakuken.or.jp (NM);
youle@helix.nih.gov (RJY)

**Abstract** Damaged mitochondria are selectively eliminated by mitophagy. Parkin and PINK1, gene products mutated in familial Parkinson's disease, play essential roles in mitophagy through ubiquitination of mitochondria. Cargo ubiquitination by E3 ubiquitin ligase Parkin is important to trigger selective autophagy. Although autophagy receptors recruit LC3-labeled autophagic membranes onto damaged mitochondria, how other essential autophagy units such as ATG9A-integrated vesicles are recruited remains unclear. Here, using mammalian cultured cells, we demonstrate that RABGEF1, the upstream factor of the endosomal Rab GTPase cascade, is recruited to damaged mitochondria via ubiquitin binding downstream of Parkin. RABGEF1 directs the downstream Rab proteins, RAB5 and RAB7A, to damaged mitochondria, whose associations are further regulated by mitochondrial Rab-GAPs. Furthermore, depletion of RAB7A inhibited ATG9A vesicle assembly and subsequent encapsulation of the mitochondria by autophagic membranes. These results strongly suggest that endosomal Rab cycles on damaged mitochondria are a crucial regulator of mitophagy through assembling ATG9A vesicles.
DOI: https://doi.org/10.7554/eLife.31326.001

## Introduction

To maintain cell homeostasis, damaged mitochondria have to be eliminated from cells, especially post-mitotic cells such as neurons. Parkin and PINK1, two gene products mutated in familial Parkinson's disease, function in a mitochondrial quality control pathway via autophagy, called mitophagy (*Pickrell and Youle, 2015*; *Yamano et al., 2016*). PINK1 is an ubiquitin kinase (*Kane et al., 2014*; *Kazlauskaite et al., 2014*; *Koyano et al., 2014*) that is rapidly degraded through the N-end rule pathway after translocation into healthy mitochondria (*Yamano and Youle, 2013*) but accumulates on damaged mitochondria upon loss of membrane potential (*Geisler et al., 2010*; *Matsuda et al., 2010*; *Narendra et al., 2010*; *Vives-Bauza et al., 2010*). Recent studies have revealed that phosphorylated ubiquitin made by PINK1 on damaged mitochondria functions as a receptor for E3

ubiquitin ligase Parkin (*Okatsu et al., 2015*; *Ordureau et al., 2015*; *Ordureau et al., 2014*) as well as an activator of the ligase activity. Once Parkin is recruited to damaged mitochondria, it becomes fully active through further phosphorylation (*Kondapalli et al., 2012*; *Shiba-Fukushima et al., 2012*). As a result, through positive feedback ubiquitination cycles (*Okatsu et al., 2015*; *Ordureau et al., 2014*), many outer mitochondrial membrane (OMM) proteins including MFN1, MFN2 and TOMM20 are ubiquitinated by Parkin (*Bingol et al., 2014*; *Rose et al., 2016*; *Sarraf et al., 2013*), and some of these ubiquitinated substrates either undergo proteasomal degradation (*Tanaka et al., 2010*; *Yoshii et al., 2011*) or behave as signals that trigger autophagy-dependent lysosomal degradation (*Narendra et al., 2008*).

Macroautophagy (hereafter referred to as autophagy) is a eukaryotic conserved mechanism for mediating degradation of cellular components. Many autophagy proteins have been identified in mammals, and form several functional units (*Feng et al., 2014*; *Mizushima et al., 2011*). The most upstream regulator is the ULK1 complex, which consists of ULK1, ATG13, RB1CC1/FIP200, and ATG101. Another regulator, the phosphatidylinositol 3-kinase (PI3K) complex generates phosphati-dylinositol 3-phosphate (PI3P). ATG9A is the only known multispannning membrane protein among the essential autophagy proteins. Under basal conditions, ATG9A localizes on the trans-Golgi net-work (TGN) and recycling endosomes as well as in the cytosol as small membrane vesicles (*Reggiori and Tooze, 2012*; *Yamamoto et al., 2012*). Upon autophagy stimulation, ATG9A is tran-siently recruited to the sites of the autophagosome biogenesis, although its precise role remains unclear. Two ubiquitin-like conjugation machineries, ATG5-ATG12 and phosphatidylethanolamine (PE)-LC3 are also important for making enclosed autophagosomes (*Tsuboyama et al., 2016*). PE-LC3 (also known as lipidated LC3) stably associates with autophagic membranes, which can be used as an autophagy marker (*Kabeya et al., 2000*). Furthermore, factors of cellular membrane trafficking systems including small GTPases such as Rab proteins and their regulators, and SNARE proteins have recently shown to support autophagy biogenesis.

Accumulating evidence has shown that autophagy coordinates the ubiquitin/proteasome system, especially in a selective autophagy including mitophagy. The well-characterized proteins that con-nect autophagy and ubiquitin are autophagy receptors such as OPTN, CALCOCO2/NDP52, and SQSTM1/p62, all of which contain both ubiquitin-binding domains (UBDs) and LC3-interacting motifs (*Wild et al., 2014*). Therefore, these proteins can deliver LC3-labeled phagophore to ubiquitin-deco-rated mitochondria. However, other essential autophagy units also have to be recruited onto the sur-face of the damaged mitochondria, probably independent of LC3 (*Itakura et al., 2012*), to elongate phagophore and/or construct enclosed autophagosomes, while our understanding is limited.

We previously showed that the mitochondrial protein FIS1 and its interacting mitochondrial Rab GTPase-activating proteins (GAPs), TBC1D15 and TBC1D17, are important for autophagosomal bio-genesis during mitophagy (*Yamano et al., 2014*). The loss of FIS1 inhibits elimination of damaged mitochondria in cultured cells (*Yamano et al., 2014*), inhibits elimination of paternal mitochondria in mouse embryos (*Rojansky et al., 2016*) and accumulates LC3 in *C. elegans* (*Shen et al., 2014*). The obvious phenomenon during Parkin-mediated mitophagy following loss of FIS1 or TBC1D15 in cul-tured cells is the accumulation of LC3B that is suppressed by *RAB7A* siRNA. Therefore, accumulation of excess amounts of LC3B during mitophagy is RAB7-dependent, but the molecular mechanism remains unclear.

In this study, we show that RABGEF1, a guanine nucleotide exchange factor (GEF) of endosomal Rab proteins, which contains UBDs, is recruited to damaged mitochondria in a Parkin-dependent manner. RABGEF1 directs the downstream Rab proteins, RAB5 and RAB7A, to damaged mitochon-dria, that is further regulated by mitochondrial Rab-GAPs. Depletion of RAB7A inhibits ATG9A vesi-cle assembly and subsequent encapsulation of the mitochondria by autophagic membranes. From these results, we propose that endosomal Rab cycling at damaged mitochondria is a crucial regula-tor of mitophagy through recruitment of ATG9A vesicles.

## Results

### RAB7A is recruited to damaged mitochondria during mitophagy

We first explored the localization of RAB7A during mitophagy. Without stimulation of mitophagy, Parkin localizes throughout the cytosol and YFP-RAB7A mainly colocalizes with a late endosome/

lysosome marker LAMP2 in WT, *FIS1-/-* and *TBC1D15/17* double knockout (DKO) cells (*Figure 1—figure supplement 1A*), but not with a mitochondrial marker TOMM20 (*Figure 1—figure supplement 1B*), confirming that RAB7A is on the late endosomes and lysosomes under basal conditions. Three hours of valinomycin treatment, which disrupts the membrane potential, resulted in Parkin translocation to mitochondria (*Figure 1A*). Although a minute amount of YFP-RAB7A colocalizes with damaged mitochondria in WT cells, loss of mitochondrial Rab-GAPs, TBC1D15 and TBC1D17, or loss of their receptor FIS1 greatly induced colocalization of YFP-RAB7A with the mitochondria (*Figure 1A*). Although substantial amounts of RAB7A still localize on the lysosomes even in *FIS1-/-* and *TBC1D15/17* DKO cells during mitophagy (*Figure 1B*), RAB7A recruitment to mitochondria does not reflect lysosome localization with mitochondria (*Figure 1C*). Next, we observed localization of endogenous RAB7A during mitophagy. We first confirmed that immunostained signals of anti-RAB7 antibody we used were overlapped with LAMP2 signals (*Figure 1—figure supplement 2A*) and that RAB7A siRNA treatment drastically reduced the signal (*Figure 1—figure supplement 2B*), indicating that the antibody specifically recognizes endogenous RAB7A. When mitophagy was induced by 3 hr of valinomycin treatment, a portion of endogenous RAB7A was colocalized with damaged mitochondria in *FIS1-/-* and *TBC1D15/17* DKO cells (*Figure 1—figure supplement 2C*). These results demonstrate that both endogenous and exogenous RAB7A, but not the lysosome, is recruited to mitochondria during mitophagy and RAB7A accumulation is enhanced by loss of mitochondrial Rab-GAPs, consistent with the idea that Rab-GAPs facilitate the dissociation of Rab proteins from target membranes. We next asked how RAB7A localizes to mitochondria. Without Parkin overexpression, RAB7A was not recruited to mitochondria, indicating that mitophagy stimulation by Parkin is required (*Figure 1D*). In general, membrane tethering of Rab proteins requires their GTPase activities and C-terminal posttranslational prenylation (*Müller and Goody, 2017*). Therefore, we tested the RAB7A T22N mutant that locks the protein in a GDP-bound conformation, thereby reducing the GTPase activity and a C-terminal 4aa deletion (ΔC) mutant in which prenylation is impaired. Although localization patterns of RAB7A Q67L (a GTP-locked) and T22N mutants under growing conditions looked similar to that of RAB7A WT, following mitophagy induction, the T22N mutant was minimally recruited to mitochondria (*Figure 1E and G*) in contrast to the Q67L mutant (*Figure 1E*). Furthermore, the ΔC mutant did not translocate to mitochondria when Parkin was recruited to damaged mitochondria (*Figure 1F and G*). These data indicate that the GTPase activity and the C-terminal prenylation are essential for RAB7A to be recruited to damaged mitochondria.

To clarify whether RAB7A associates to the OMM directly or via autophagic membranes, we first tested colocalization of RAB7A and LC3B-labeled autophagic membranes. Under basal conditions, only a small number of YFP-LC3B dots was detected (*Figure 2A*), but after valinomycin treatment, WT cells formed many dot-like structures near mitochondria, indicating autophagic membranes (*Figure 2B*, panel a and b). Interestingly, RAB7A was found to be associated with WT mitochondria weakly, and sometimes enriched portions of the RAB7A signal on the mitochondria overlapped YFP-LC3B signal, suggesting that RAB7A may interact with both autophagic membranes and damaged mitochondria. On the other hand, loss of FIS1 or loss of TBC1D15/17 caused excessive YFP-LC3B accumulation as reported previously (*Yamano et al., 2014*). 2HA-RAB7A strongly associates with damaged mitochondria in areas lacking YFP-LC3B-labeled membranes in *FIS1-/-* and *TBC1D15/17* DKO cells (*Figure 2B*, panel c-f), suggesting direct association of RAB7A to the mitochondria. To rule out the possibility that other membrane structures intervene between RAB7A and the OMM for the association, we conducted immunoelectron microscopy (*Figure 2C*). While gold particles attached to YFP-RAB7A mainly localize on late endosomes/lysosomes in *TBC1D15/17* DKO cells under basal conditions (*Figure 2C,a and b*), they directly associate to the surface of the OMM after mitophagy stimulation with no other membrane found between RAB7A and the OMM (*Figure 2C,c–f*). Therefore, it appears that RAB7A can directly attach to the OMM.

## RAB7A is required for an early step of autophagosome biogenesis

It has been reported that RAB7A is involved in the fusion between autophagosomes and lysosomes at the downstream stage of autophagy (*Gutierrez et al., 2004*; *Jäger et al., 2004*). However, since (1) RAB7A is recruited to mitochondria during mitophagy and (2) mitochondrial RAB7A-GAPs control precise encapsulation of mitochondria by autophagic membranes during mitophagy (*Yamano et al., 2014*), we hypothesized that RAB7A acts more upstream in autophagosome biogenesis during mitophagy. To examine autophagic structures in RAB7A-depleted cells, we knocked down RAB7A

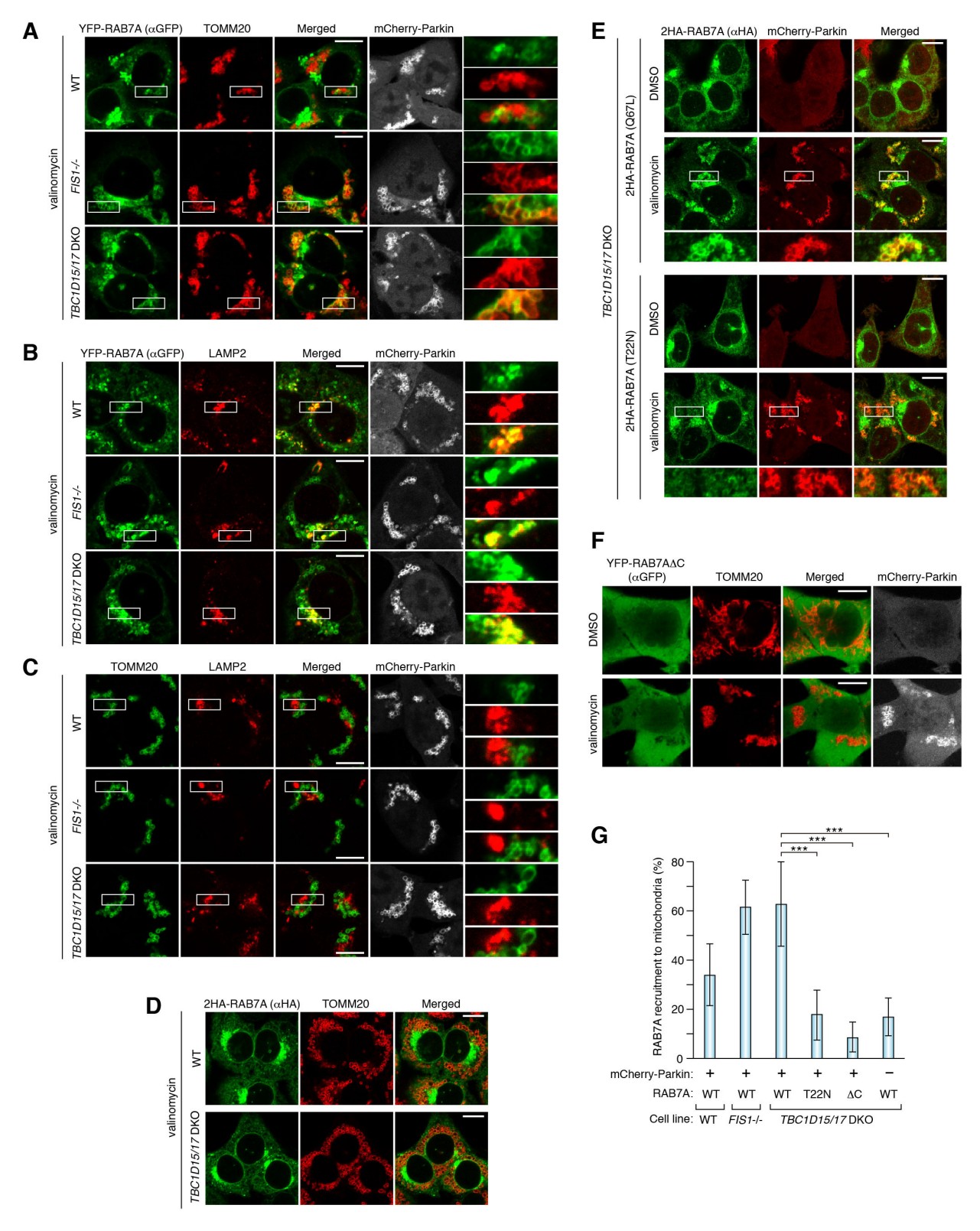

**Figure 1.** RAB7A is recruited to damaged mitochondria during mitophagy. The indicated HCT116 cells stably expressing mCherry-Parkin and YFP-RAB7A (**A and B**), mCherry-Parkin alone (**C**), 2HA-RAB7A alone (**D**), or mCherry-Parkin and indicated RAB7A mutant (**E and F**) were treated with DMSO or valinomycin for 3 hr followed by immunostaining. Magnified images are also shown for A-C, and E. Bars, 10 μm. (**G**) Quantification of RAB7A recruitment to damaged mitochondria. Overlapped RAB7A signals with TOMM20 per total RAB7A signals were measured. Total RAB7A signal in each

*Figure 1 continued on next page*

*Figure 1 continued*

cell set to 100%. Error bars represent mean ± SE of at least two independent experiments. Statistical differences were determined by one-way ANOVA with Dunnett's multiple comparisons test. ***p<0.001.

DOI: https://doi.org/10.7554/eLife.31326.002

The following source data and figure supplements are available for figure 1:

**Source data 1.** This excel file contains quantification of recruitment of RAB7 WT and mutants to damaged mitochondria.

DOI: https://doi.org/10.7554/eLife.31326.005

**Figure supplement 1.** RAB7A localization under basal conditions.

DOI: https://doi.org/10.7554/eLife.31326.003

**Figure supplement 2.** Mitochondrial recruitment of endogenous RAB7A during mitophagy.

DOI: https://doi.org/10.7554/eLife.31326.004

(*Figure 3A*) and observed distributions of various autophagy-related proteins required for phagophore and/or autophagosome formation at different stages; ULK1 and ATG13 for early nucleation, DFCP1 (also known as ZFYVE1) as an omegasome marker, and WIPI1 that binds to PI3P, and ATG16L1 required for lipidation of LC3. They all formed dot-like or ring-like structures near damaged mitochondria after 3 hr of mitophagy stimulation (*Figure 3—figure supplement 1*), but the numbers or morphologies of them did not detectably change between control and *RAB7A* siRNA-treated cells (*Figure 3—figure supplement 1*). When YFP-LC3B was observed during mitophagy, many spherical structures surrounding the mitochondrial matrix protein PDHA1 were found in control siRNA-treated cells (*Figure 3B*). However, while YFP-LC3B was still recruited to damaged mitochondria as dot-like structures during mitophagy in *RAB7A* siRNA-treated cells, the number of spherical structures containing PDHA1 was significantly reduced (*Figure 3B and C*). These results suggest that the phagophores in RAB7A-depleted cells cannot efficiently engulf damaged mitochondria. Interestingly, we found that the recruitment of ATG9A was also impaired by depletion of RAB7A. ATG9A is present in the cytosol as well as on the TGN in small vesicles (*Figure 3D*, and [*Puri et al., 2013*; *Young et al., 2006*]). After 3 hr of mitophagy stimulation, endogenous ATG9A assembled into larger dots on mitochondria in control siRNA-treated cells (*Figure 3D*), but much less so after knocking down of RAB7A (*Figure 3D and E*). We also noticed that ATG9A on the Golgi apparatus was reduced upon mitophagy in control siRNA cells, but not in *RAB7A* siRNA-treated cells (*Figure 3D and F*). Furthermore, most of the ATG9A dots colocalized with LC3-labeled autophagic membranes upon mitophagy stimulation (*Figure 4A*). To analyze this in more detail, we used *TBC1D15/17* DKO cells because loss of the mitochondrial RAB7-GAPs excessively activates RAB7A and enhances LC3B accumulation near damaged mitochondria. As shown in *Figure 4B and C* and *Figure 4—figure supplement 1A*, higher levels of ATG9A vesicles were localized close to the LC3-labeled structures in *TBC1D15/17* DKO cells. In sharp contrast, the number of ATG16L1 dots in *TBC1D15/17* DKO cells was quite similar to that in WT cells (*Figure 4—figure supplement 1B*), suggesting the specificity of ATG9A for expanding the phagophore.

## RAB7A is required for elimination of damaged mitochondria

To assess whether RAB7A is required for mitophagy, we performed mitochondria clearance assays. HeLa cells stably expressing YFP-Parkin were treated with control or *RAB7A* siRNA followed by valinomycin exposure for various times (*Figure 5A*). MFN2 was completely degraded within 3 hr (*Figure 5A*), consistent with the previous reports (*Tanaka et al., 2010*; *Yoshii et al., 2011*) showing that MFN2 is rapidly degraded by ubiquitin-proteasome pathway prior to autophagic degradation. In control siRNA cells, TOMM20 and MT-CO2 (a mitochondrial DNA (mtDNA)-encoded matrix protein) were efficiently degraded after 36 hr of valinomycin treatment (*Figure 5A*). However, in *RAB7A* siRNA-treated cells, protein levels of TOMM20 and especially MT-CO2 were not altered even after 36 hr (*Figure 5A*). A defect in degradation of lipidated LC3B was also observed in RAB7A-depleted cells (*Figure 5A*), suggesting that autophagic flux is inhibited by a reduced level of RAB7A. Similar mitophagy defects were also observed by microscopic analysis (*Figure 5B*). Parkin translocation in *RAB7A* siRNA-treated cells was found to be similar to those in control siRNA-treated cells (*Figure 5C*). 24 hr of valinomycin treatment resulted in efficient degradation of TOMM20, PDHA1, and mtDNA in control siRNA-treated cells, whereas *RAB7A* siRNA treatment blocked their

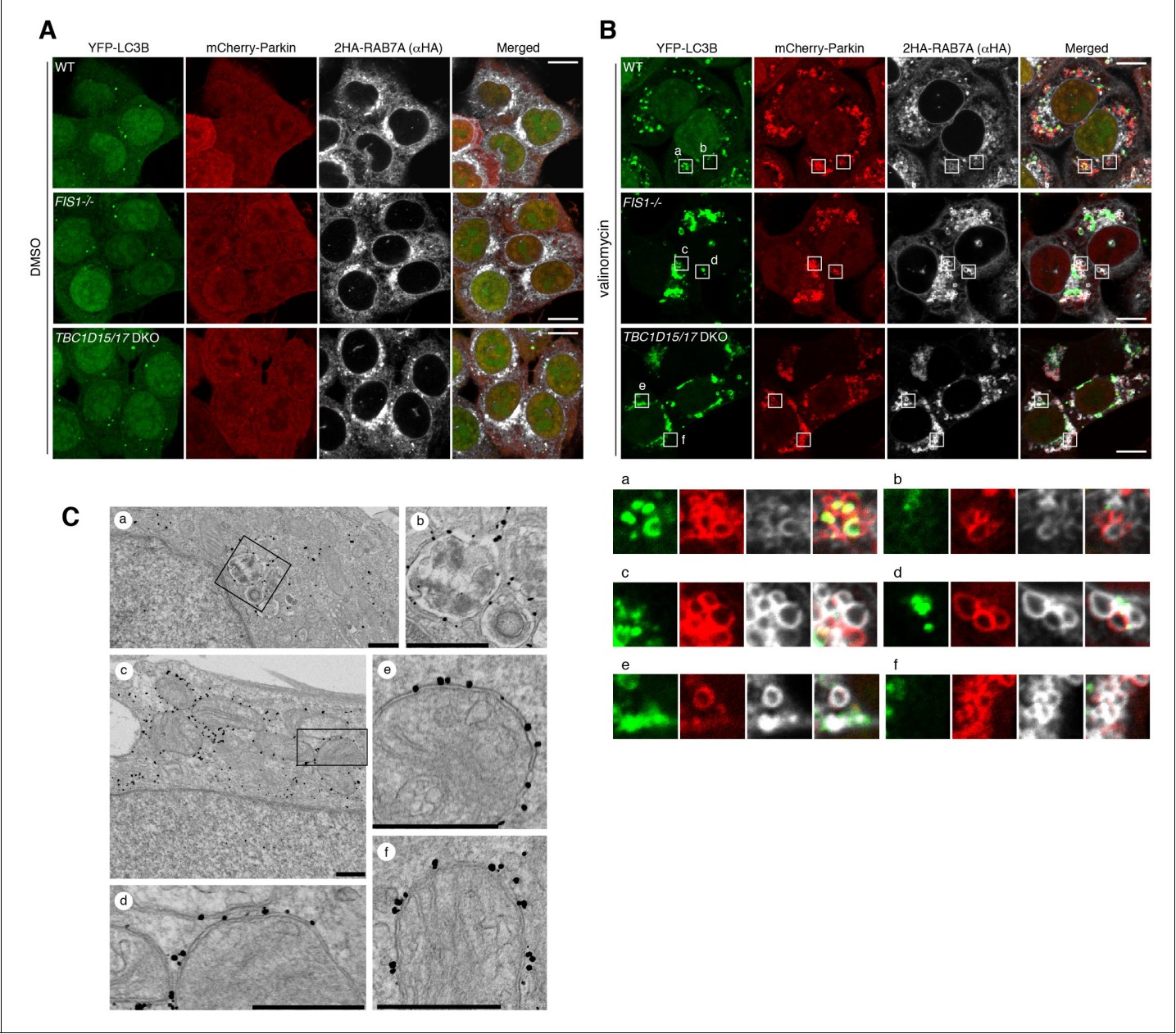

**Figure 2.** RAB7A directly associates to the outer membrane of damaged mitochondria. (**A and B**) The indicated HCT116 cells stably expressing YFP-LC3B, mCherry-Parkin and 2HA-RAB7A were treated with DMSO (**A**) or valinomycin (**B**) for 3 hr, and subjected to immunostaining. The magnified images of the cells treated with valinomycin were shown in a-f. Bars, 10 μm. (**C**) *TBC1D15/17* DKO cells stably expressing mCherry-Parkin and YFP-RAB7A were treated with DMSO (a and b) or valinomycin (c–f) for 3 hr and then subjected to immunoelectron microscopy with anti-GFP antibody. Panels b and d are the magnified images of boxes in panels a and c, respectively. Bars, 500 nm.
DOI: https://doi.org/10.7554/eLife.31326.006

degradation (*Figure 5B and D–F*). These results indicate that RAB7A is important for efficient mitochondrial clearance.

## MON1/CCZ1 complex directs RAB7A to damaged mitochondria

RAB7A normally associates with the late endosome/lysosome membranes. To clarify how RAB7A is recruited to damaged mitochondria, we identified RAB7A-interacting proteins during mitophagy by mass spectrometry. We used the RAB7A GDP-locked T22N mutant as (1) Rab-GEFs are

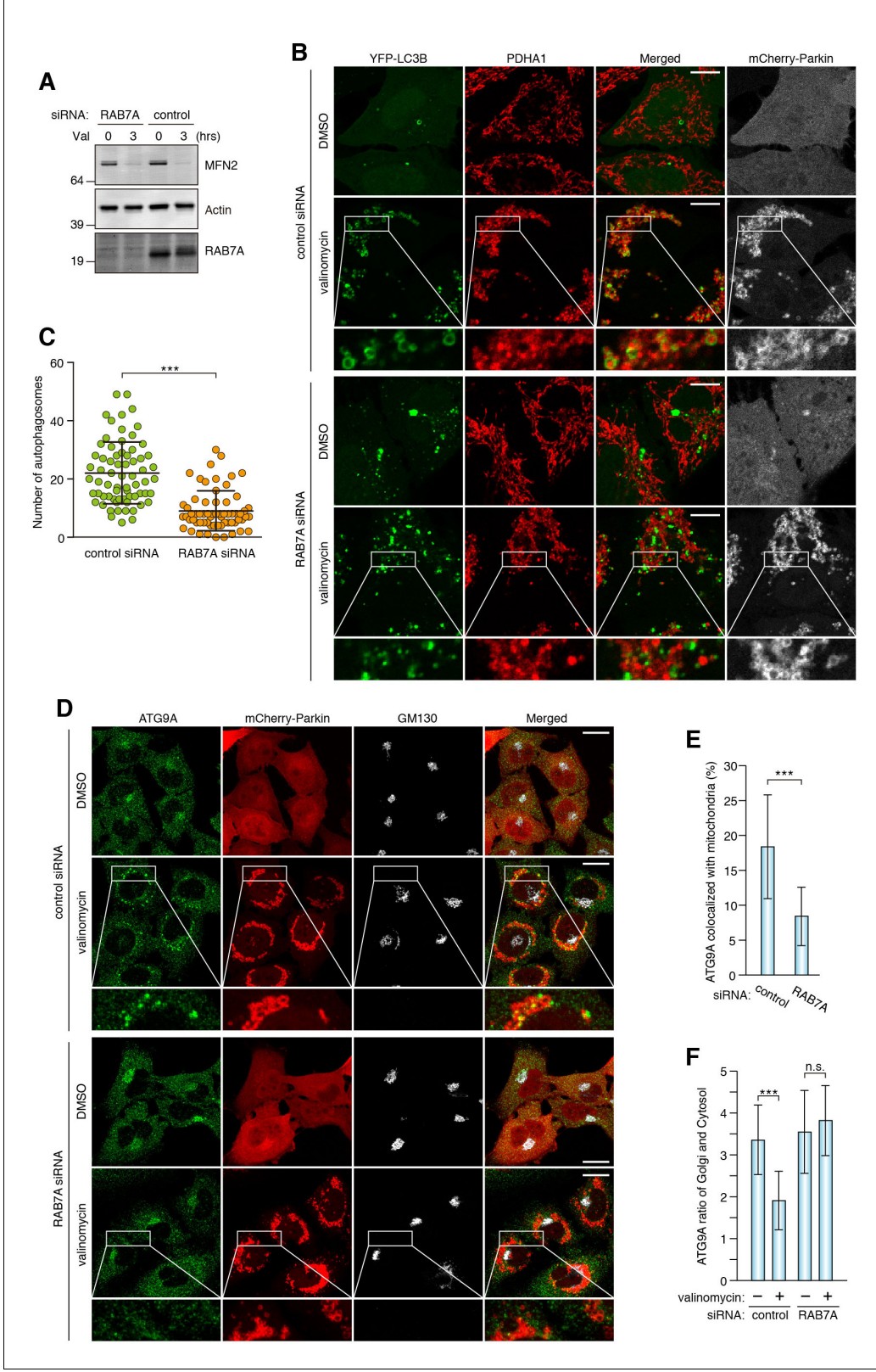

**Figure 3.** RAB7A is required for ATG9A recruitment to damaged mitochondria and encapsulation by autophagic membranes. (**A**) siRNA-treated HeLa cells stably expressing mCherry-Parkin were treated with DMSO or valinomycin (Val) for 3 hr. Total cell lysates were analyzed by immunoblotting. (**B**) siRNA-treated HeLa cells stably expressing mCherry-Parkin and YFP-LC3B were treated with DMSO or valinomycin for 3 hr. The fixed cells were subjected to immunostaining. Images are displayed as z-stacks of five confocal slices. The magnified pictures of the cells treated with valinomycin were

*Figure 3 continued on next page*

*Figure 3 continued*

shown. Bars, 10 μm. (C) The number of autophagosomes containing PDHA1 inside in each cell was counted. Error bars represent mean ±SE of at least two independent experiments. Statistical differences were determined by student's t-test. ***p<0.001. (D) The fixed cells as in (A) were subjected to immunostaining. Images are displayed as z-stacks of five confocal slices. Magnified images are shown for cells treated with valinomycin. Bars, 20 μm. (E) Quantification of ATG9A recruitment to damaged mitochondria in (D). Overlapped ATG9A signals with mitochondria-localized mCherry-Parkin per total ATG9A signals were measured. Total ATG9A signal in each cell set to 100%. Error bars represent mean ±SE. Cells from at least two independent experiments were quantified. Statistical difference was determined by student's t-test. ***p<0.001. (F) Quantification of ATG9A localization on Golgi apparatus (see the Materials and methods for the detail). Error bars represent mean ±SE. Cells from at least two independent experiments were quantified. Statistical difference was determined by student's t-test ***p<0.001; n.s., not significant.

DOI: https://doi.org/10.7554/eLife.31326.007

The following source data and figure supplements are available for figure 3:

**Source data 1.** The number of autophagosomes during mitophagy in cells treated with control or RAB7A siRNA.

DOI: https://doi.org/10.7554/eLife.31326.009

**Source data 2.** This excel file contains quantification of ATG9A recruitment to damaged mitochondria.

DOI: https://doi.org/10.7554/eLife.31326.010

**Source data 3.** Quantification of ATG9A localization.

DOI: https://doi.org/10.7554/eLife.31326.011

**Figure supplement 1.** Recruitment of autophagy-related proteins to mitochondria during mitophagy.

DOI: https://doi.org/10.7554/eLife.31326.008

**Figure supplement 1—source data 1.** source data1 This excel file contains quantification of recruitment/assembly of autophagy-related proteins, GFP-ULK1, GFP-ATG13, GFP-DFCP1, GFP-WIPI1 and ATG16L1 during mitophagy.

DOI: https://doi.org/10.7554/eLife.31326.012

preferentially bound to Rab GDP-locked form and (2) Rab-GEFs might be a determinant for the corresponding Rab membrane targeting (*Blümer et al., 2013*). 2 × HA-tagged RAB7A T22N mutant (2HA-RAB7A T22N) and mCherry-Parkin stably expressing cells treated with or without valinomycin were subjected to HA-agarose immunoprecipitation (HA-IP) followed by mass spectrometry analysis. Several proteins identified by our mass spectrometry are previously known to be involved in Rab and lysosomal functions (*Figure 6A*, and *Supplementary file 1*); ARL8B for lysosome dynamics (*Marwaha et al., 2017*; *Mrakovic et al., 2012*), ATP6V1A, a component of lysosomal ATPase, GDI1 and GDI2 that can solubilize prenylated Rab proteins in the cytosol, MON1A, MON1B and CCZ1, subunits of the MON1/CCZ1 complex as a RAB7A-GEF (*Gerondopoulos et al., 2012*; *Nordmann et al., 2010*). Although several mitochondrial proteins were identified as RAB7A-interacting proteins (*Figure 6A*), we further analyzed the MON1/CCZ1 complex. We first confirmed that the MON1/CCZ1 complex preferentially interacts with the GDP form of RAB7A. Although overexpressed MON1/CCZ1 complex remained in the cytosol in the presence of DsRed-RAB7A WT (*Figure 6B*), it accumulated on late endosomes/lysosomes when expressing DsRed-RAB7A T22N (*Figure 6B*). To test whether the MON1/CCZ1 complex is involved in RAB7A recruitment to damaged mitochondria, we knocked them down in *TBC1D15/17* DKO cells because of the efficient RAB7A mitochondrial translocation (*Figure 6C*). Although knocking down either MON1A or MON1B did not inhibit RAB7A recruitment to the mitochondria, the combination of *MON1A* and *MON1B* siRNAs impaired RAB7A recruitment (*Figure 6C–E*). This is because MON1A and MON1B function redundantly in the cells we used (*Figure 6A*). *CCZ1* siRNA also inhibited RAB7A recruitment to the mitochondria (*Figure 6C–E*), indicating that the MON1/CCZ1 complex is required for mitochondrial recruitment of RAB7A.

## RAB5 is recruited to damaged mitochondria during mitophagy

In the endocytosis pathway, RAB5 functions as the upstream Rab protein, which transfers the signal to the downstream MON1/CCZ1 complex. Therefore, we examined RAB5 localization during mitophagy. Under basal conditions, 3 × HA-tagged mouse RAB5C (3HA-mRAB5C) mainly localizes on the early endosomes marked by EEA1 (*Figure 7—figure supplement 1A*) and does not colocalize with mitochondria (*Figure 7A*). On the other hand, a part of the 3HA-mRAB5C signals in WT cells weakly merged with TOMM20 after 3 hr of valinomycin treatment. Furthermore, RAB5C recruitment to damaged mitochondria was greatly enhanced in *TBC1D15/17* DKO cells (*Figure 7A*). Another RAB5 variant RAB5B behaved similarly to RAB5C (*Figure 7—figure supplement 1B and C*).

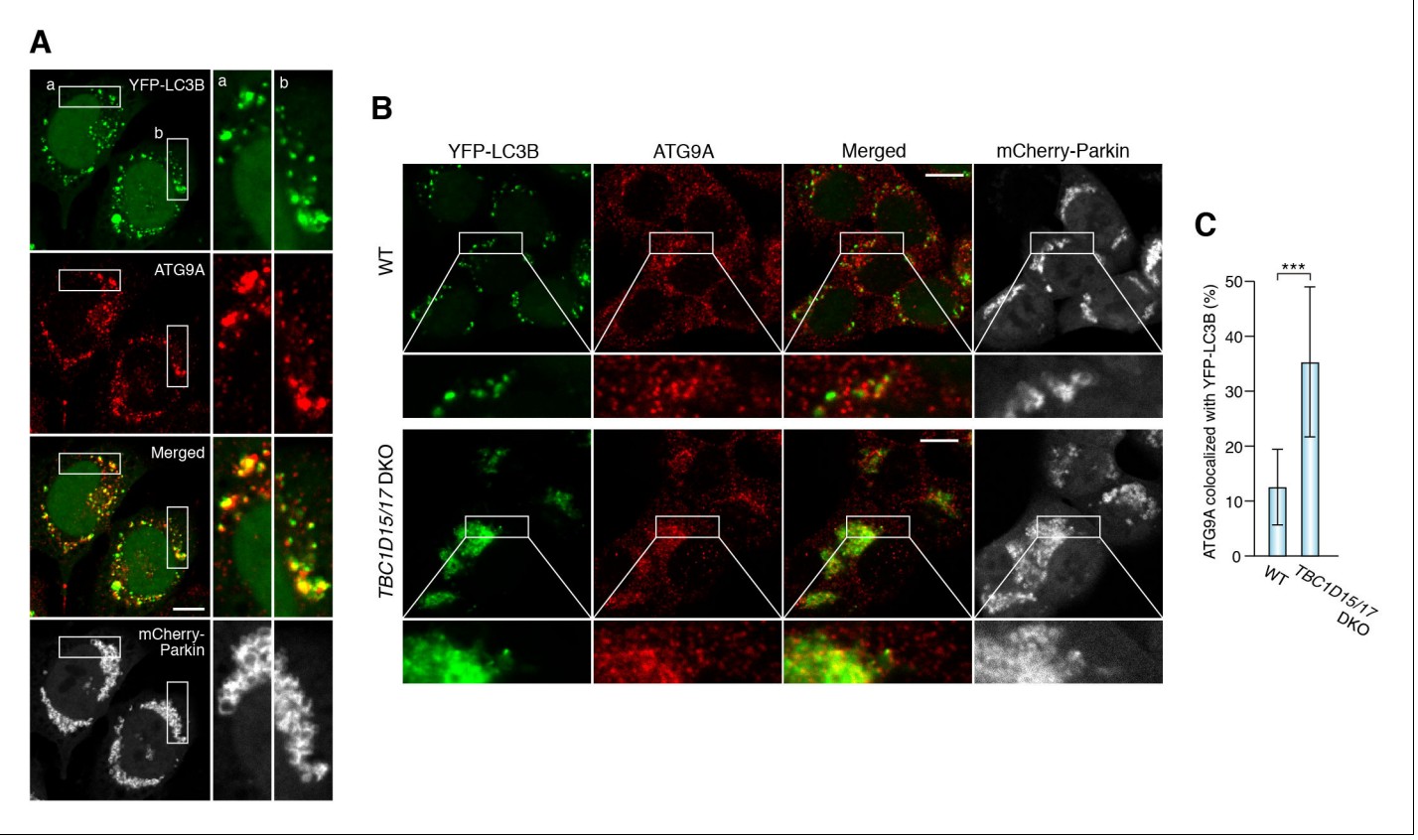

**Figure 4.** Loss of mitochondrial Rab-GAPs induced excess amounts of ATG9A on damaged mitochondria. (**A**) HeLa cells stably expressing YFP-LC3B and mCherry-Parkin were treated with valinomycin for 3 hr followed by immunostaining. Magnified images of boxes (**a**) and (**b**) are shown in the right. Bars, 10 µm. (**B**) The indicated HCT116 cells stably expressing YFP-LC3B and mCherry-Parkin were treated with valinomycin for 3 hr followed by immunostaining. Bars, 10 µm. (**C**) YFP-LC3B and ATG9A signals prepared as in (**B**) were processed, and overlapped ATG9A signal with YFP-LC3B per total ATG9A signals in each cell were measured. Total ATG9A signal in each cell set to 100%. Error bars represent mean ±SE. Statistical difference was determined by student's t-test. ***$p < 0.001$.

DOI: https://doi.org/10.7554/eLife.31326.013

The following source data and figure supplement are available for figure 4:

**Source data 1.** Quantification of colocalization of ATG9A and YFP-LC3B signals during mitophagy.
DOI: https://doi.org/10.7554/eLife.31326.015

**Figure supplement 1.** ATG9A and ATG16L1 recruitment to mitochondria in *TBC1D15/17* DKO cells.
DOI: https://doi.org/10.7554/eLife.31326.014

Endogenous RAB5, which localizes on the early endosomes under normal growing conditions (*Figure 7B*), was also recruited to damaged mitochondria during mitophagy and the recruitment was accelerated by loss of TBC1D15/17 or loss of FIS1 (*Figure 7C*). In sharp contrast, RAB17 and RAB29 stayed in the cytosol and on the Golgi apparatus, respectively, during mitophagy (*Figure 7— figure supplement 1D–F*), demonstrating that mitochondrial recruitment is specific for RAB5 and RAB7A.

## RABGEF1 is recruited to damaged mitochondria in a ubiquitin-binding-dependent manner

We examined the role of the RAB5-GEF, RABGEF1, and found that mitochondrial recruitment of RAB5C was inhibited by knocking down of RABGEF1 but not by knocking down of the MON1/CCZ1 complex or RAB7A (*Figure 7D and E*). Interestingly, in addition to the GEF domain, RABGEF1 has two different UBDs at the N-terminus (*Lee et al., 2006*; *Penengo et al., 2006*). Overexpressed GFP-tagged mouse RABGEF1 (GFP-mRABGEF1) localizes in the cytosol under normal growing conditions (*Figure 8A and B*). With Parkin translocation after 3 hr of valinomycin treatment, GFP-mRABGEF1 is

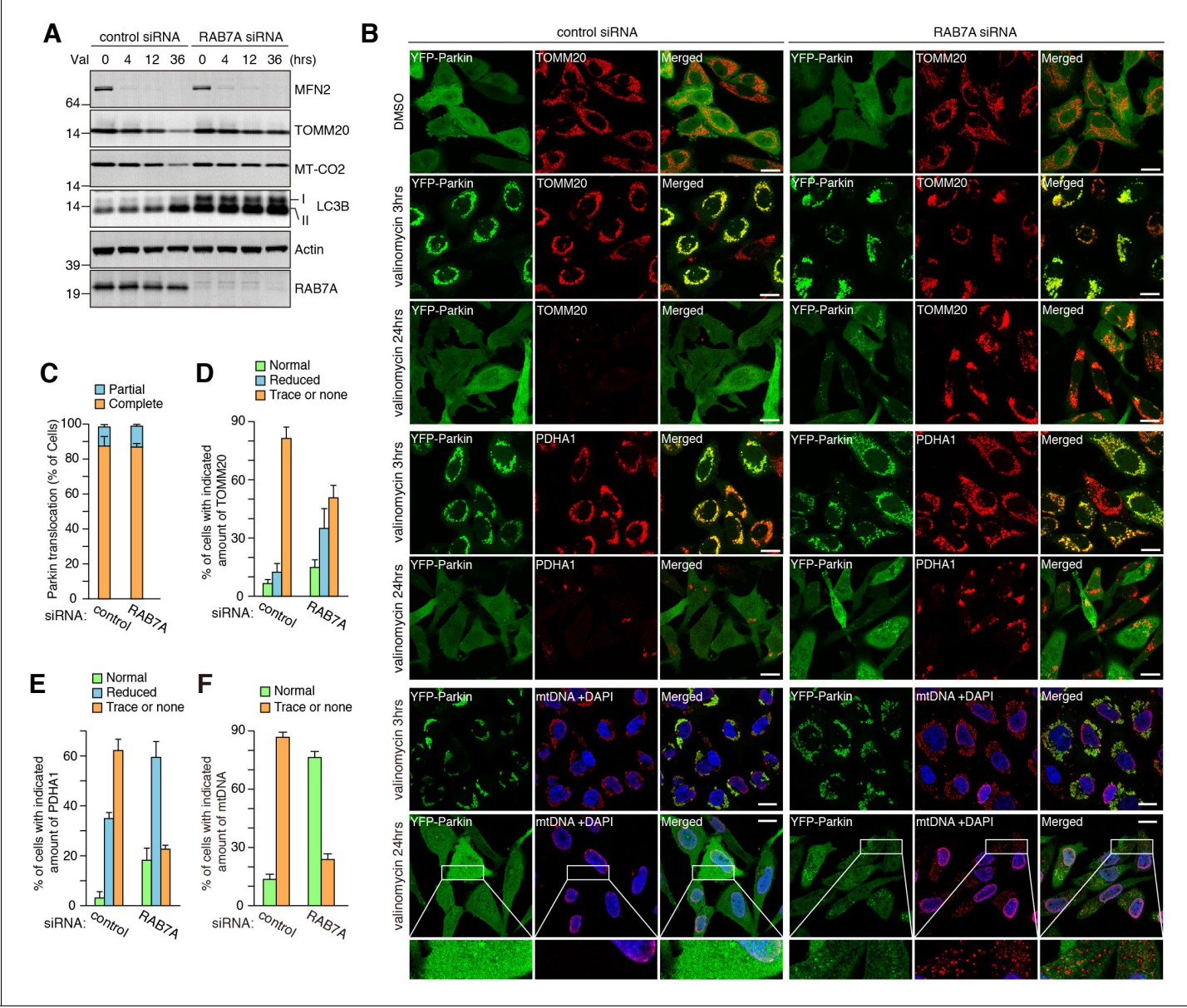

**Figure 5.** RAB7A is required for mitophagy. (**A**) siRNA-treated HeLa cells stably expressing YFP-Parkin were treated with valinomycin (Val) for the indicated times and total cell lysates were analyzed by immunoblotting. I and II denote unmodified and lipidated LC3B, respectively. (**B**) Cells in (**A**) were subjected to immunostaining. DAPI was used for nuclei staining since anti-mtDNA antibody non-specifically stains nuclei of the cells having no mtDNA. Magnified pictures were shown for mtDNA degradation in cells treated with valinomycin for 24 hr. Bars, 20 μm. (**C**) Quantification of YFP-Parkin translocation to mitochondria after 3 hr of valinomycin treatment. Partial and complete denote that Parkin translocates to some of or all mitochondria, respectively. Error bars represent mean ±SE and over 100 cells were counted in each of three separate wells. (D - F) Percentages of cells having the indicated amount of TOMM20 (**D**), PDHA1 (**E**) and mtDNA (**F**) after 24 hr of valinomycin treatment were shown. Error bars represent mean ±SE from three independent replicates. Over 100 cells were counted in each of three separate wells.

DOI: https://doi.org/10.7554/eLife.31326.016

The following source data is available for figure 5:

**Source data 1.** This excel file contains quantification of YFP-Parkin recruitment to damaged mitochondria, degradation of TOMM20, and degradation of PDHA1 upon mitophagy.

DOI: https://doi.org/10.7554/eLife.31326.017

**Source data 2.** Quantification of mtDNA degradation upon mitophagy.

DOI: https://doi.org/10.7554/eLife.31326.018

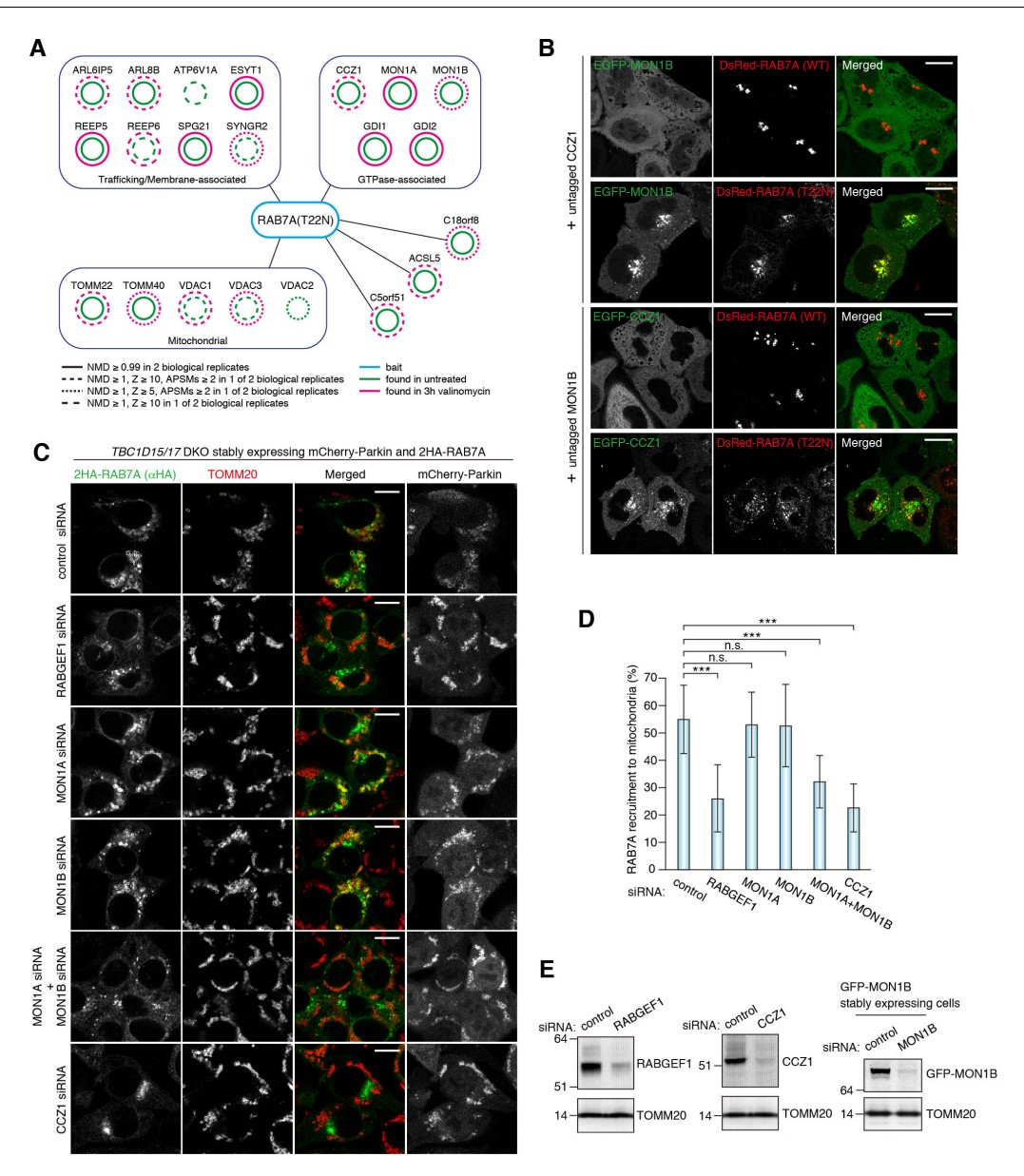

**Figure 6.** MON1/CCZ1 complex is required for RAB7A recruitment to damaged mitochondria. (**A**) Lysates of *TBC1D15/17* DKO HCT116 cells stably expressing mCherry-Parkin and 2HA-RAB7A (T22N) were subjected to HA-IP, followed by trypsin digestion and mass spectrometric analysis. High-confidence candidate interacting protein (HCIPs) partners of RAB7A (T22N) are color-coded: untreated (green outline) and 3 hr of valinomycin (magenta outline). Line quality as described in figure key indicates criteria used for inclusion. (**B**) EGFP-MON1B and untagged CCZ1 (or EGFP-CCZ1 and untagged MON1B) were transiently expressed with DsRed-RAB7A (WT or T22N) in HeLa cells. Bars, 10 μm. (**C**) siRNA-treated *TBC1D15/17* DKO cells stably expressing mCherry-Parkin and 2HA-RAB7A were treated with valinomycin for 3 hr and subjected to immunostaining. Bars, 10 μm. (**D**) RAB7A recruitment to mitochondria in (**C**) was quantified. Total signals of 2HA-RAB7A in each cell set to 100%. Error bars represent mean ±SE of at least two independent experiments. Statistical differences were determined by one-way ANOVA with Dunnett's multiple comparisons test. ***p<0.001; n.s., not significant. (**E**) HCT116 cells or those stably expressing GFP-MON1B were treated with the indicated siRNAs. Total cell lysates were analyzed by immunoblotting. GFP-MON1B was detected by anti-GFP antibody.

DOI: https://doi.org/10.7554/eLife.31326.019

The following source data is available for figure 6:

**Source data 1.** This excel file contains quantification of 2HA-RAB7A recruitment to mitochondria in *TBC1D15/17* DKO cells.

DOI: https://doi.org/10.7554/eLife.31326.020

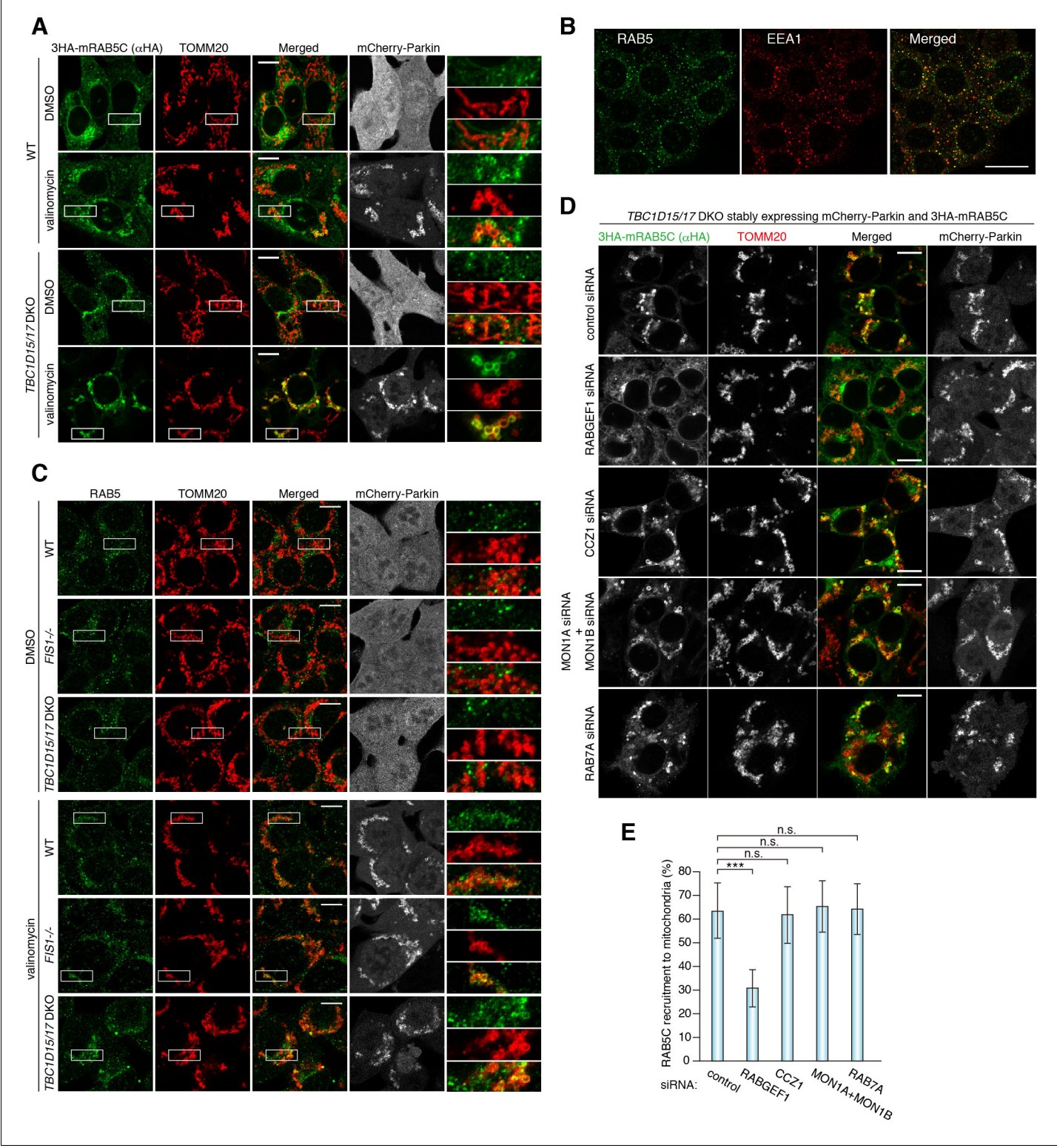

**Figure 7.** RAB5 is recruited to damaged mitochondria during mitophagy. (**A**) WT or *TBC1D15/17* DKO cells stably expressing mCherry-Parkin and 3HA-mRAB5C were treated with DMSO or valinomycin for 3 hr. The cells were subjected to immunostaining. The magnified pictures were shown in the right. Bars, 10 μm. (**B**) HCT116 cells were subjected to immunostaining. Bars, 20 μm. (**C**) The indicated HCT116 cells stably expressing mCherry-Parkin were treated with DMSO or valinomycin for 3 hr followed by immunostaining. The magnified pictures were shown in the right. Bars, 10 μm. (**D**) siRNA-treated *TBC1D15/17* DKO cells stably expressing mCherry-Parkin and 3HA-mRAB5C were treated with valinomycin for 3 hr and subjected to immunostaining. Bars, 10 μm. (**E**) Quantification of mRAB5C recruitment to damaged mitochondria in (**D**). Total signals of 3HA-mRAB5C in each cell set to 100%. Error

*Figure 7 continued on next page*

*Figure 7 continued*

bars represent mean ±SE of at least two independent experiments. Statistical differences were determined by one-way ANOVA with Dunnett's multiple comparisons test. ***p<0.001; n.s., not significant.

DOI: https://doi.org/10.7554/eLife.31326.021

The following source data and figure supplement are available for figure 7:

**Source data 1.** This excel file contains quantification of 3HA-RAB5C recruitment to mitochondria in *TBC1D15/17* DKO cells.

DOI: https://doi.org/10.7554/eLife.31326.023

**Figure supplement 1.** Localization of RAB5B, RAB29, and RAB17 during mitophagy.

DOI: https://doi.org/10.7554/eLife.31326.022

also robustly recruited to the mitochondria (*Figure 8A and B*). RABGEF1 recruitment to mitochondria requires both Parkin expression and loss of membrane potential (*Figure 8—figure supplement 1A*) and was observed both in WT and *TBC1D15/17* DKO cells with similar efficiencies (*Figure 8— figure supplement 1A*). GFP-mRABGEF1 expressed at lower level was also recruited to the mitochondria (*Figure 8—figure supplement 1B and C*). When introducing Y26A/A58D mutations to disrupt ubiquitin-binding ability (described later and [*Lee et al., 2006*; *Penengo et al., 2006*]), the translocation of RABGEF1 was completely blocked (*Figure 8A and C*), strongly suggesting that RABGEF1 binds to poly-ubiquitin chains that Parkin builds on damaged mitochondria. Consistent with this, knocking down of the downstream Rab-related factors including MON1, CCZ1, and RAB7A did not affect the mitochondrial recruitment of RABGEF1 (*Figure 8—figure supplement 2*).

To investigate biochemical features of RABGEF1 UBD, we purified recombinant mouse RABGEF1 UBD (WT, Y26A, A56D and Y26A/A58D) fused with a GST tag at the N-terminus (GST-mRABGEF1) from bacterial cells (*Figure 8—figure supplement 3A*). Recombinant linear tandem ($1\times$, $2\times$, $3\times$, and $4\times$) ubiquitins (*Figure 8—figure supplement 3B*) were incubated with GST-mRABGEF1. GST-mRABGEF1 WT efficiently pulled down $2\times$ and higher ubiquitin chains (*Figure 8D and E*). When single mutations, Y26A or A58D, were introduced, the amounts of eluted ubiquitin were slightly reduced compared to WT (*Figure 8D and E*). In sharp contrast, the double mutation Y26A/A58D completely lost the ability to bind ubiquitin (*Figure 8D and E*). The binding ability to phosphorylated ubiquitin, which was produced by incubating with recombinant TcPINK1 (*Figure 8—figure supplement 3C*) was also tested. The overall binding profiles were quite similar to those of non-phosphorylated ubiquitin (*Figure 8D and E*). As Parkin on damaged mitochondria mainly makes K48- and K63-linked ubiquitin chains (*Ordureau et al., 2014*), we built K48- and K63-linked ubiquitin chains in vitro (*Figure 8F*). RABGEF1 WT, and single Y26A and A58D mutants could bind both K48- and K63-linked ubiquitin or phosphorylated ubiquitin chains, but Y26A/A58D mutant completely did not (*Figure 8F*). To calculate the binding constant between ubiquitin and RABGEF1, isothermal titration calorimetry (ITC) was used. ITC showed that mouse RABGEF1 UBD interacts with monomeric non-phosphorylated and phosphorylated ubiquitin with a dissociation constant of 2.2 μM and 5.6 μM, respectively (*Figure 8G*, and *Figure 8—figure supplement 3D and E*). All these biochemical results strongly suggest that neither ubiquitin-chain linkage nor S65 phosphorylation substantially affect the binding affinity between ubiquitin and RABGEF1.

## RABGEF1 is important for efficient elimination of damaged mitochondria

The above results suggest that RABGEF1 is involved in mitophagy. To degrade endogenous RABGEF1 rapidly, we utilized auxin-inducible degron technology (*Nishimura et al., 2009*). The mAID (a 68-aa fragment of the original AID/IAA17) tag was genetically inserted at the C-terminus of RABGEF1 in HCT116 cells stably expressing an F-box component of plant E3 ligase, OsTIR1 (See Materials and methods for the details, (*Natsume et al., 2016*), yielding the *RABGEF1-mAID* cell line. RABGEF1-mAID, but not WT RABGEF1, was degraded within 16 hr of indolacetic acid (IAA) treatment (*Figure 9A*), a half-life that is much shorter than conventional siRNA methods. To test whether RABGEF1 is required for mitochondrial elimination via autophagy, WT and *RABGEF1-mAID* cells stably expressing YFP-Parkin were treated with IAA and valinomycin for various times. The efficiency of Parkin translocation (*Figure 9B*) and MFN2 degradation (*Figure 9C*) did not change after RABGEF1 degradation. However, TOMM20 and MT-CO2 degradation was slightly inhibited in *RABGEF1-mAID*

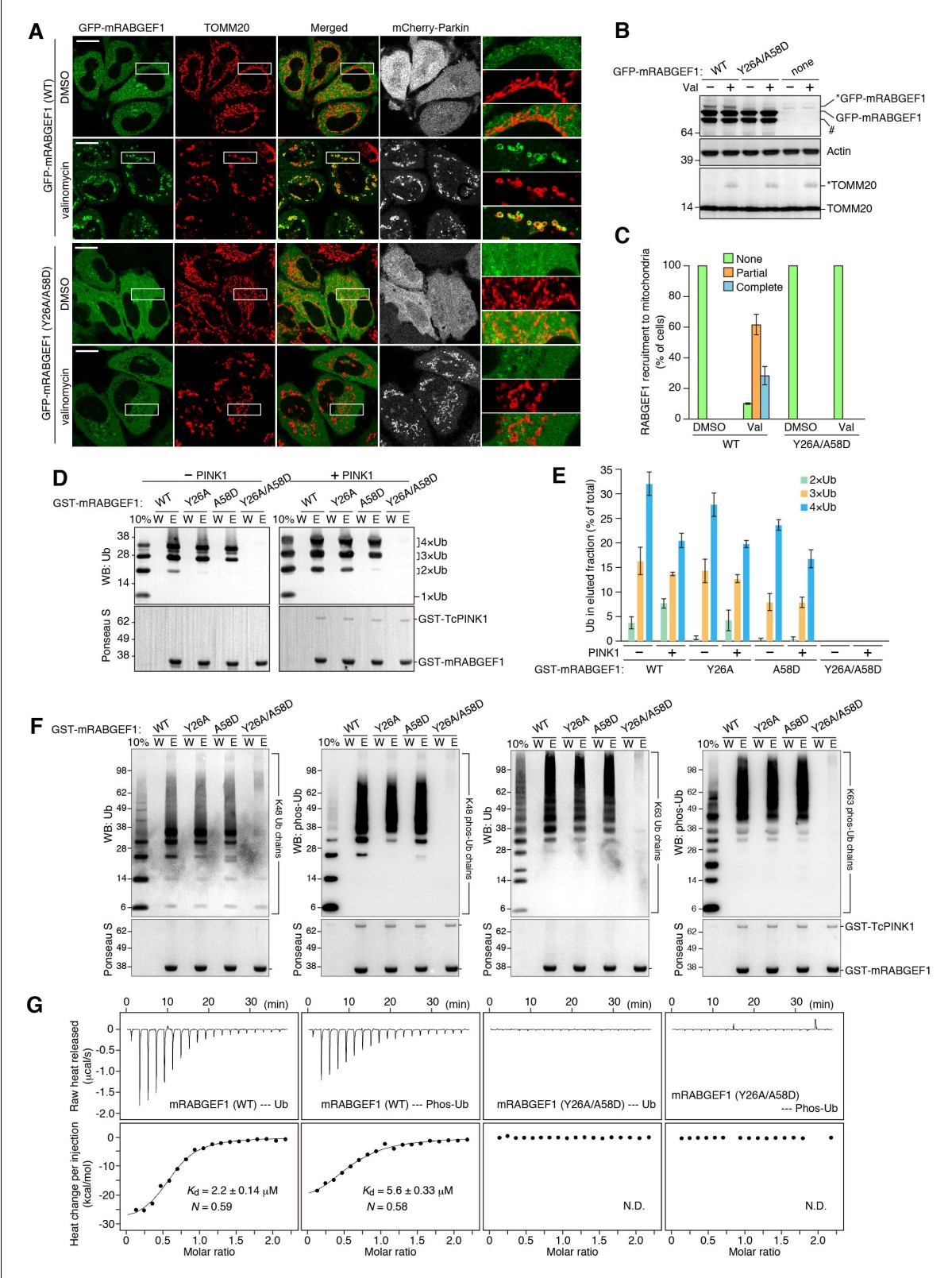

**Figure 8.** RABGEF1 is recruited to the damaged mitochondria in a ubiquitin-binding dependent manner. (**A**) HeLa cells transiently expressing mChery-Parkin and GFP-mRABGEF1 were treated with DMSO or valinomycin for 3 hr followed by immunostaining. The magnified pictures were shown in the right. Bars, 10 μm. (**B**) Total cell lysates of (**A**) were analyzed by immunoblotting. Anti-GFP antibody was used for the GFP-mRABGEF1 detection. * and # denote ubiquitinated forms and truncated forms, respectively. (**C**) Quantification of RABGEF1 recruitment to damaged mitochondria in (**A**). None,

*Figure 8 continued on next page*

*Figure 8 continued*

partial and complete denote that GFP-mRABGEF1 signals were overlapped with no, some of, and all mitochondria, respectively. (D) Recombinant ubiquitin (Ub) pre-treated with or without GST-TcPINK1 was subjected to pull-down assay with GST-mRABGEF1. W and E indicate wash and eluted fractions, respectively. 10%, 10% of input. (E) Percentages of the amount of ubiquitin in the eluted fraction in (D) were shown. The error bars represent mean ±SE from three independent experiments. (F) K48-linked and K63-linked Ub chains pre-treated with or without GST-TcPINK1 were subjected to pull-down assay with GST-mRABGEF1. (G) Interactions between GST-mRABGEF1 (WT or Y26A/A58D) and ubiquitin or phosphorylated ubiquitin were measured by ITC. N, stoichiometry of binding.

DOI: https://doi.org/10.7554/eLife.31326.024

The following source data and figure supplements are available for figure 8:

**Source data 1.** Quantification of RABGEF1 recruitment to damaged mitochondria during mitophagy.
DOI: https://doi.org/10.7554/eLife.31326.028
**Source data 2.** Binding affinities of recombinant GST-mRABGEF1 with ubiquitin or phosphorylated ubiquitin.
DOI: https://doi.org/10.7554/eLife.31326.029
**Source data 3.** Binding affinities of recombinant GST-mRABGEF1 with ubiquitin or phosphorylated ubiquitin.
DOI: https://doi.org/10.7554/eLife.31326.030
**Figure supplement 1.** RABGEF1 recruitment to mitochondria during mitophagy.
DOI: https://doi.org/10.7554/eLife.31326.025
**Figure supplement 1—source data 1.** This excel file contains quantification of RABGEF1 (WT and Y26A/A58D mutant) recruitment to mitochondria in HCT116 (WT and *TBC1D15/17* DKO) cells.
DOI: https://doi.org/10.7554/eLife.31326.031
**Figure supplement 2.** Mitochondrial recruitment of RABGEF1 is not affected by the downstream Rabs and Rab-related factors.
DOI: https://doi.org/10.7554/eLife.31326.026
**Figure supplement 2—source data 2.** Quantification of RABGEF1 recruitment to mitochondria in HeLa cells treated with the indicated siRNA during mitophagy.
DOI: https://doi.org/10.7554/eLife.31326.032
**Figure supplement 2—source data 3.** Quantification of RABGEF1 recruitment to mitochondria in HCT116 cells treated with the indicated siRNA during mitophagy.
DOI: https://doi.org/10.7554/eLife.31326.033
**Figure supplement 3.** Preparation of recombinant RABGEF1 and ubiquitin.
DOI: https://doi.org/10.7554/eLife.31326.027

cells when compared to WT cells (*Figure 9C*). We also applied a more quantitative and sensitive mitophagy assay: Mitochondria-targeted mKeima (mt-mKeima) FACS assay (*Lazarou et al., 2015*). When damaged mitochondria are engulfed into lysosomes, a spectral shift of mt-mKeima occurs owing to the low pH following lysosomal fusion (*Katayama et al., 2011*). We found a substantial fluorescent shift by 6 hr of OAQ (oligomycin and antimycin A) treatment in WT cells, which was impaired in *RABGEF1-mAID* cells (*Figure 9D and E*). These results strongly suggest that RABGEF1 is important for efficient elimination of damaged mitochondria through autophagy.

## Mitochondrial localization of TBC1D15 and TBC1D17

TBC1D15 was identified as a FIS1-binding protein (*Onoue et al., 2013*), and we previously confirmed that mitochondrial localization of TBC1D15 and TBC1D17 depends on FIS1 (*Yamano et al., 2014*) (). As previously reported, overexpressed TBC1D15 and TBC1D17 localize in cytosol, but FIS1 overexpression direct them to the mitochondria (*Figure 10A*). To test whether endosomal Rab proteins and their GEFs regulate the mitochondrial localization of TBC1D15 during mitophagy, we knocked them down and observed endogenous TBC1D15 during mitophagy. In control siRNA-treated cells, endogenous TBC1D15 was colocalized with TOMM20, and none of the endosomal Rab knock-down abrogated mitochondrial localization of TBC1D15 (*Figure 10B*), strongly suggesting that mitochondrial localization of TBC1D15 (and TBC1D17) depends on FIS1, but not endosomal Rabs and their related factors.

## Discussion

Mitochondria, organelles believed to have evolved from proteobacteria, are quite independent from endomembranes. As mitochondria obtain their constituents (protein, tRNA and lipid) by individual transport systems, they were thought to have little contact with vesicle transport systems including

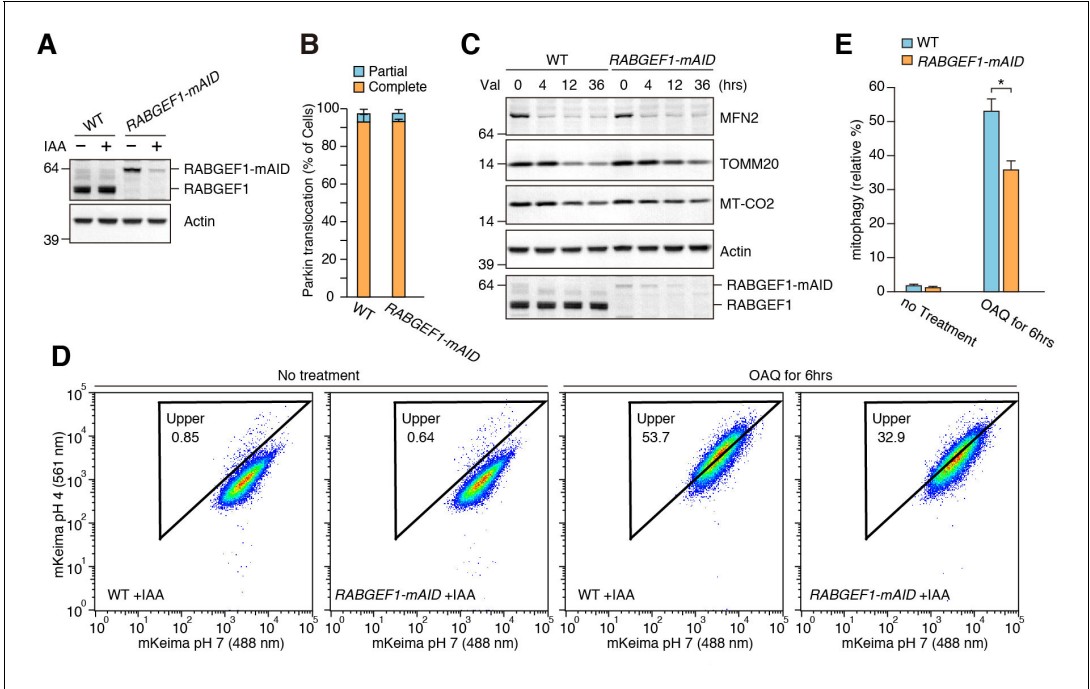

**Figure 9.** RABGEF1 is important for mitochondrial clearance. (**A**) WT and *RABGEF1-mAID* HCT116 cells were treated with or without IAA for 16 hr. Total cell lysates were analyzed by immunoblotting. (**B**) Quantification of Parkin recruitment to mitochondria in WT and *RABGEF1-mAID* HCT116 cells after 3 hr of valinomycin treatment. Partial and complete denote that YFP-Parkin signals were overlapped with some of and all mitochondria, respectively. (**C**) YFP-Parkin stably expressing WT and *RABGEF1-mAID* HCT116 cells pre-treated with IAA were treated with valinomycin for the indicated times. Total cell lysates were analyzed by immunoblotting. (**D**) WT and *RABGEF1-mAID* HCT116 cells stably expressing YFP-Parkin and mt-mKeima were treated with IAA for 16 hr followed by DMSO or OAQ for 6 hr and subjected to FACS analysis. Plots are representative of n = 3 experiments. (**E**) Quantification of mitophagy in (**D**). Error bars represent mean ±SE of three independent experiments. Statistical differences were determined by student's t-test. *p<0.05.

DOI: https://doi.org/10.7554/eLife.31326.034

The following source data is available for figure 9:

**Source data 1.** Quantification of YFP-Parkin recruitment to mitochondria in RABGEF1-mAID HCT116 and the corresponding WT cells during mitophagy.
DOI: https://doi.org/10.7554/eLife.31326.035
**Source data 2.** Quantification of mitophagy using mt-mKeima and FACS analysis.
DOI: https://doi.org/10.7554/eLife.31326.036

Rab protein-regulated pathways. However, in this study, we found that core regulators of vesicle transport system in the endosomal Rab cascade have an ability to associate with mitochondria during their engulfment by autophagosomes (*Figure 11*). In the endocytosis pathway, sequential transport from early to late endosomes requires evolutionarily conserved Rab cascade in the correct order; RABGEF1 (RAB5-GEF), RAB5, MON1/CCZ1 (RAB7-GEF) and then RAB7A.

Taking advantage of mitochondrial RAB7-GAPs (TBC1D15 and TBC1D17) KO cells, we found RAB5 and RAB7A were associated to damaged mitochondria during Parkin-mediated mitophagy. Mitochondrial recruitment of RAB5 and RAB7A was impaired by knocking down of RABGEF1, while knocking down of MON1/CCZ1 complex impaired only RAB7A. We also found that two elements for membrane tethering, proper GTPase activity and prenylation are required for the mitochondrial targeting of RAB7A. Therefore, the sequential Rab cascade on damaged mitochondrial membranes is in the same order as that occurring during endocytosis. Moreover, RABGEF1, the most upstream GEF for the endosomal Rab cascade, recognizes mitochondrial damage though the UBD (*Figure 11*). According to recent accumulating evidence, ubiquitination of damaged mitochondria constitutes a signal for encapsulation by autophagic membranes (*Herhaus and Dikic, 2015*). Poly-ubiquitinated chains conjugated to many different OMM proteins are recognized by UBD-containing proteins. The most studied ubiquitin-binding proteins are autophagy receptors that also contains LC3-interacting motif (*Birgisdottir et al., 2013*) indicating that they recruit LC3-labeled membranes to the damaged

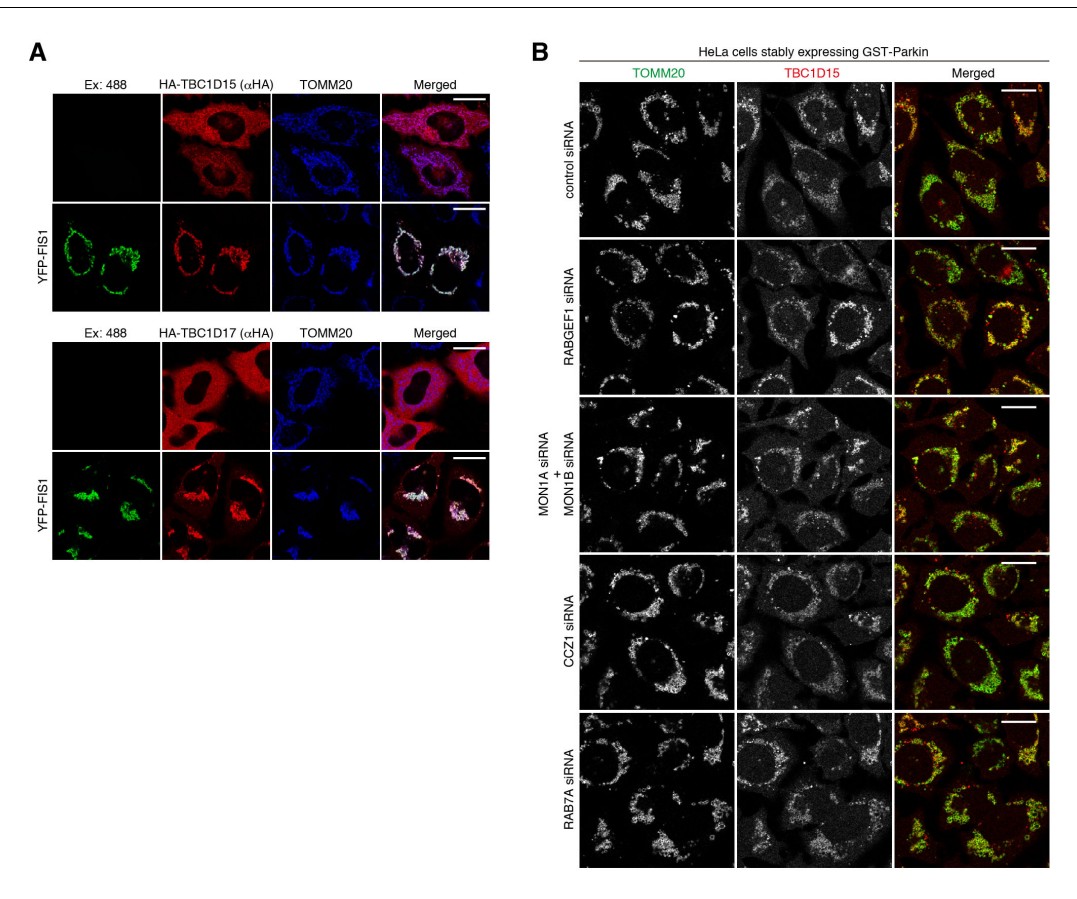

**Figure 10.** Mitochondrial localization of TBC1D15. (**A**) HA-TBC1D15 (upper) and HA-TBC1D17 (lower) with or without YFP-FIS1 were transiently expressed in HeLa cells. The cells were subjected to immunostaining. Bars, 20 μm. (**B**) HeLa cells stably expressing GST-Parkin were treated with the indicated siRNA. After 3 hr of valinomycin treatment, cells were subjected to immunostaining. Bars, 10 μm.
DOI: https://doi.org/10.7554/eLife.31326.037

mitochondria. In this study, we identified that RABGEF1 is recruited to damaged mitochondria in a UBD-dependent manner, and found that mitochondrial RABGEF1 is a platform for transfer of the signal to the downstream Rab cascade. A previous report showed that experimentally inducing RAB-GEF1 localization to mitochondrial membranes would recruit RAB5 to the mitochondria, in support of our results (*Blümer et al., 2013*).

What does RAB7A do on the damaged mitochondria? siRNA-based experiment strongly suggests that RAB7A directs ATG9A vesicles to damaged mitochondria and/or facilitates the assembly of ATG9A vesicles near damaged mitochondria during Parkin-mediated mitophagy. Moreover, when excess amounts of RAB7A are recruited to damaged mitochondria by deletion of mitochondrial Rab-GAPs, an excess amount of ATG9A vesicles is also recruited along with LC3-labeled autophagic membranes. Considering the correlation between the amount of RAB7A associated to the damaged mitochondria and the efficiency of ATG9A recruitment, mitochondria-targeted RAB7A likely regulates ATG9A vesicle trafficking during mitophagy (*Figure 11*). Since ATG9A is a multispanning membrane protein integrated in small vesicles, it is expected that ATG9A moves along the vesicle trafficking network. Indeed, ATG9A localizes to different membrane compartments such as TGN, recycling membrane and plasma membrane, whose trafficking as well as autophagosome formation under starvation conditions are regulated by several Rab proteins (RAB1 and RAB11) and their regulators (TBC1D5, TBC1D14, and TRAPPIII complex) (*Lamb et al., 2016*; *Longatti et al., 2012*; *Popovic and Dikic, 2014*). Here, we identified RAB7A as a new mitophagy-specific ATG9A vesicle regulator. ATG9A might be required not only qualitatively but also quantitatively for expanding LC3-labeled membrane structures. Given that Rab proteins function in membrane fusion, RAB7A on

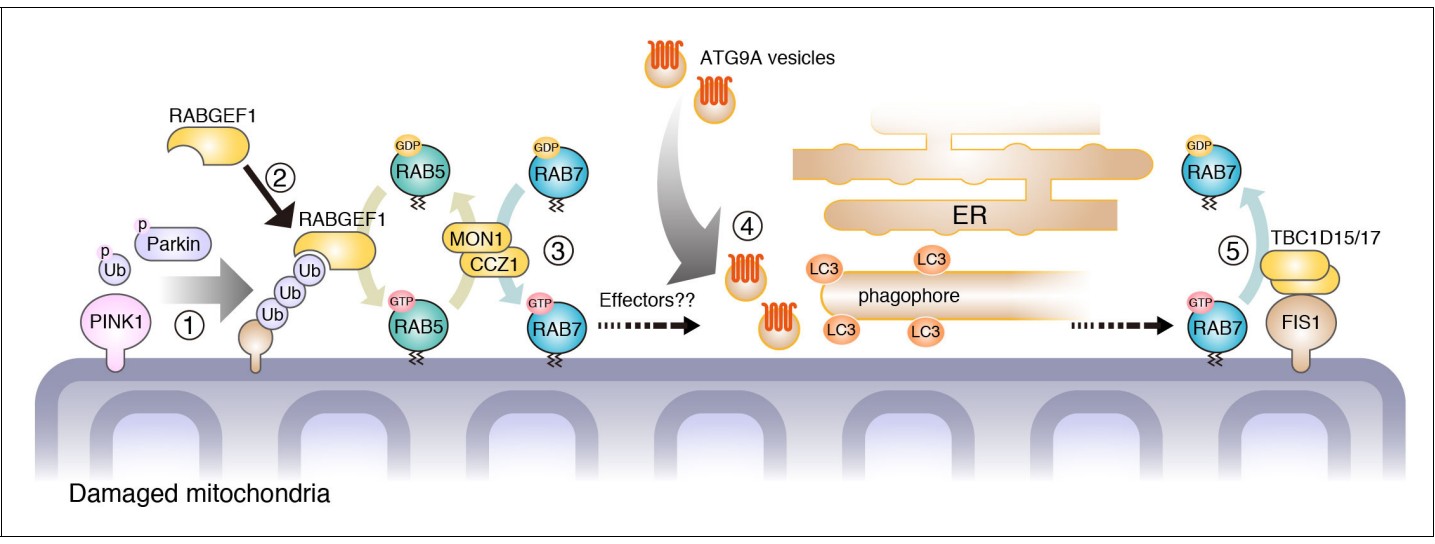

**Figure 11.** Proposed model of mitophagy regulated by endosomal Rab cycles. (1) Through phosphorylation by PINK1, Parkin and ubiquitin ubiquitinate damaged mitochondria. (2) RABGEF is recruited to mitochondria and (3) endosomal Rab cycles including RAB5 and MON1/CCZ1 complex direct RAB7A to the mitochondria. (4) ATG9A vesicles are recruited to the autophagosome formation sites, in a RAB7A-dependent manner, where ATG9A vesicles and LC3-labeled autophagic membranes are assembled. (5) Mitochondrial Rab-GAPs, TBC1D15 and TBC1D17, dissociate RAB7A from the mitochondrial membranes to complete the Rab cycles.

DOI: https://doi.org/10.7554/eLife.31326.038

mitochondria may facilitate the expansion of phagophore by assisting fusion of ATG9A vesicles with the phagophore. During starvation-induced autophagy, ATG9A is hard to detect on the growing phagophore (*Koyama-Honda et al., 2013*) probably because they do not need to stay there to enlarge autophagosomal structures. We, therefore, propose that upon Parkin-mediated mitophagy, RAB7A regulates coordinated action of making precise autophagosome between LC3-labeled pre-autophagosome membrane and ATG9A vesicles.

The fact that RAB7A appears on the mitochondrial surface more so in *TBC1D15/17* DKO cells may reflect that mitochondria represent an off target site for RAB7A and that the mitochondrial RAB7-GAPs dissociate RAB7A so it may target adjacent phagophore membranes. Located at junctions between the growing phagophore and mitochondrial membranes, RAB7A recruits ATG9A-bound vesicles presumably to foster phagophore expansion. RAB5, on the other hand, appears targeted directly to the OMM by RABGEF1 via binding ubiquitin chains catalyzed there by Parkin. One scenario is that RAB5 on the mitochondria recruits RAB7A to this site where it is dissociated by TBC1D15/17, perhaps to target local growing phagophores and augment their growth around mitochondria. As OPTN and NDP52 initiate phagophore recruitment to invading bacteria, it is tempting to suggest that a similar Rab pathway may exist to expand phagophore around pathogens. RAB7A is also recruited to the growing phagophore during xenophagy (*Yamaguchi et al., 2009*). Therefore, phagophore expansion regulated by RAB7A might be a shared molecular hub between mitophagy and xenophagy that relays ubiquitin signaling to the endomembrane system for autophagosome assembly.

## Materials and methods

**Key resources table**

| Reagent type (species) or resource | Designation | Source or reference | Identifiers | Additional information |
|---|---|---|---|---|
| Cell line (*Homo sapiens*) | HeLa | ATCC | CVCL_0030 | |

*Continued on next page*

*Continued*

| Reagent type (species) or resource | Designation | Source or reference | Identifiers | Additional information |
|---|---|---|---|---|
| Cell line (*H. sapiens*) | HCT116 | ATCC | CVCL_0291 | |
| Cell line (*H. sapiens*) | *FIS1-/-* | *Otera et al. (2010)* | | |
| Cell line (*H. sapiens*) | *TBC1D15/17* DKO | *Yamano et al. (2014)* | | |
| Cell line (*H. sapiens*) | HCT116-OsTIR1 | *Natsume et al. (2016)* | | |
| Cell line (*H. sapiens*) | *RABGEF1-mAID* | this paper | | mAID sequence were inserted into HCT116-OsTIR1 cell line to produce RABGEF1-mAID. |
| Cell line (*H. sapiens*) | HEK293T | ATCC | CVCL_0063 | |
| Antibody | Rabbit anti-GFP (polyclonal) | Abcam | ab6556 AB_305564 | 1:1000 (WB), 1:1000 (IF) |
| Antibody | Mouse anti-MFN2 (monoclonal) | Abcam | ab56889 AB_2142629 | 1:500(WB) |
| Antibody | Rabbit anti-TOMM20 (polyclonal) | Santa Cruz Biotechnology | sc-11415 AB_2207533 | 1:2000 (WB), 1:1000 (IF) |
| Antibody | Rabbit anti-LC3B | Sigma | L7543 AB_796155 | 1:1000 (WB) |
| Antibody | Mouse anti-MT-CO2 (monoclonal) | Abcam | ab110258 AB_10887758 | 1:500 (WB) |
| Antibody | Mouse anti-Actin (monoclonal) | Millipore | MAB1501R AB_2223041 | 1:2000 (WB) |
| Antibody | Mouse anti-RAB7 (monoclonal) | Abcam | ab50533 AB_882241 | 1:1000 (WB) |
| Antibody | Rabbit anti-RABGEF1 (polyclonal) | NOVUS BIOLOGICALS | NBP1-49938 AB_10012128 | 1:500 (WB) |
| Antibody | Mouse anti-CCZ1 (monoclonal) | Santa Cruz Biotechnology | sc-514290 | 1:100 (WB) |
| Antibody | Mouse anti-ubiquitin (monoclonal) | Santa Cruz Biotechnology | sc-8017 AB_628423 | 1:1000 (WB) |
| Antibody | Rabbit anti-S65 phosphorylated ubiquitin | *Koyano et al. (2014)* | | 1:500 (WB) |
| Antibody | Rabbit anti-GFP (polyclonal) | Invitrogen | A-11122 AB_221569 | 1:1000 (IF) |
| Antibody | Mouse anti-GFP (monoclonal) | Invitrogen | A-11120 AB_221568 | 1:1000 (IF) 1:500 (immuno-EM) |
| Antibody | Mouse anti-TOMM20 (monoclonal) | Santa Cruz Biotechnology | sc-17764 AB_628381 | 1:200 (IF) |
| Antibody | Mouse anti-HA (monoclonal) | MBL Life science | M180-3 AB_10951811 | 1:2000 (IF) |
| Antibody | Mouse anti-HA (monoclonal) | COVANCE | MMS-101R-500 AB_10063630 | 1:500 (IF) |
| Antibody | Mouse anti-LAMP2 (monoclonal) | Santa Cruz Biotechnology | sc-18822 AB_626858 | 1:100 (IF) |
| Antibody | Mouse anti-EEA1 (monoclonal) | BD Biosciences | 610457 AB_397830 | 1:200 (IF) |
| Antibody | Mouse anti-GM130 (monoclonal) | BD Biosciences | 610822 AB_398141 | 1:1000 (IF) |

*Continued on next page*

*Continued*

| Reagent type (species) or resource | Designation | Source or reference | Identifiers | Additional information |
|---|---|---|---|---|
| Antibody | Mouse anti-pryruvate dehydrogenase E1-alpha subunit (PDHA1) (monoclonal) | Abcam | ab110334 AB_10866116 | 1:500 (IF) |
| Antibody | Mouse anti-DNA (monoclonal) | Millipore | CBL186 AB_11213573 | 1:500 (IF) |
| Antibody | Rabbit anti-RAB5 (monoclonal) | Cell Signaling Technology | 3547 AB_2300649 | 1:200 (IF) |
| Antibody | Rabbit anti-RAB7 (monoclonal) | Cell Signaling Technology | 9367 AB_1904103 | 1:100 (IF) |
| Antibody | Rabbit anti-TBC1D15 (clonal) | A kind gift from N. Ishihara, Kurume University, Japan | | 1:50 (IF) |
| Antibody | Rabbit anti-ATG9A (clonal) | A kind gift from N. Mizushima, University of Tokyo,Japan | | 1:100 (IF) |
| Antibody | Rabbit anti-ATG16L1 | A kind gift from N. Mizushima, University of Tokyo,Japan | | 1:200 (IF) |
| Antibody | Goat anti-Rabbit IgG, Alexa Fluor 488 conjugated | Thermo Fisher Scientific | A-11034 AB_2576217 | 1:500 (IF) |
| Antibody | Goat anti-Rabbit IgG, Alexa Fluor 568 conjugated | Thermo Fisher Scientific | A-11036 AB_10563566 | 1:500 (IF) |
| Antibody | Goat anti-Rabbit IgG, Alexa Fluor 647 conjugated | Thermo Fisher Scientific | A-21245 AB_2535813 | 1:500 (IF) |
| Antibody | Goat anti-Mouse IgG, Alexa Fluor 488 conjugated | Thermo Fisher Scientific | A-11029 AB_138404 | 1:500 (IF) |
| Antibody | Goat anti-Mouse IgG, Alexa Fluor 568 conjugated | Thermo Fisher Scientific | A-11031 AB_144696 | 1:500 (IF) |
| Antibody | Goat anti-Mouse IgG, Alexa Fluor 647 conjugated | Thermo Fisher Scientific | A-21236 AB_2535805 | 1:500 (IF) |
| Antibody | Nanogold-conjugated anti-mouse IgG antibody | Nanoprobes | 2002 AB_2637031 | 1:200 (IF) |
| Antibody | Anti-rabbit IgG horseradish peroxidase-linked secondary antibodies | GE Healthcare | NA934 AB_772206 | 1:5000 (WB) |
| Antibody | Anti-HA beads | Sigma-aldrich | A2095 AB_257974 | |
| Chemical compound, drug | Lpofectamine RNAiMAX | Invitrogen | Invitrogen: 13778–150 | |
| Chemical compound, drug | FuGENE6 | Promega | Promega: E2692 | |
| Chemical compound, drug | FuGENE HD | Promega | Promega: E2311 | |
| Chemical compound, drug | DMEM | Life Technologies | Life Technologies: 31053–028 | |
| Chemical compound, drug | DMEM | Sigma-aldrich | Sigma-aldrich: D5796-500ML | |
| Chemical compound, drug | Sodium pyruvate | Life Technologies | Life Technologies: 11360–070 | |
| Chemical compound, drug | Glutamine | Life Technologies | Life Technologies: 25030–081 | |
| Chemical compound, drug | GlutaMAX | Life Technologies | Life Technologies: 35050–061 | |

*Continued on next page*

*Continued*

| Reagent type (species) or resource | Designation | Source or reference | Identifiers | Additional information |
|---|---|---|---|---|
| Chemical compound, drug | Nonessential amino acids | Life Technologies | Life Technologies: 11140–050 | |
| Chemical compound, drug | McCoy's 5A | Life Technologies | Life Technologies: 16600–082 | |
| Chemical compound, drug | Polybrene | Sigma-aldrich | Sigma-aldrich: H9268 | |
| Chemical compound, drug | Valinomycin | Sigma-aldrich | Sigma-aldrich: V0627-10MG | |
| Chemical compound, drug | Oligomycin | Calbiochem | Calbiochem: 495455–10 MG | |
| Chemical compound, drug | Antimycin A | Sigma-aldrich | Sigma-aldrich: A8674-25MG | |
| Chemical compound, drug | Q-VD-OPH | SM Biochemicals | SM Biochemicals: SMPH001 | |
| Chemical compound, drug | Q-VD-OPH | Sigma-aldrich | Sigma-aldrich: SML0063-1MG | |
| Chemical compound, drug | Indole-3-acetic acid (IAA) | Wako | Wako: 090–07123 | |
| Chemical compound, drug | G418 | Sigma-aldrich | Sigma-aldrich: G8168 | |
| Chemical compound, drug | Hygromycin B | Invitrogen | Invitrogen: 10687–010 | |
| Chemical compound, drug | DAPI | Thermo Fisher Scientific | Thermo Fisher Scientific: D3571 | |
| Chemical compound, drug | Protease inhibitor cocktail | Roche | Roche: 11 873 580 001 | |
| Chemical compound, drug | Phos-tag | Wako | Wako: 304–93521 | |
| Chemical compound, drug | DTBP (dimethyl 3,3'-dithiobispropionimidate) | Pierce | Pierce: 20665 | |
| Chemical compound, drug | TCEP (Tris(2-carboxylethyl) phosphine) | Sigma-aldrich | Sigma-aldrich: C4706-10G | |
| Chemical compound, drug | GSH (L-glutathione reduced) | Sigma-aldrich | Sigma-aldrich: G4251-25G | |
| Chemical compound, drug | PhosSTOP phosphatase inhibitor cocktail | Roche | Roche: 04 906 845 001 | |
| Chemical assay or kit | BCIP-NBT solution kit | Nacalai Tesque | Nacalai Tesque: 03937–60 | |
| Chemical assay or kit | Western Lightning Plus-ECL | PerkinElmer | PerkinElmer: NEL105001EA | |
| Peptide, recombinant protein | HA peptide | Sigma-aldrich | Sigma-aldrich: I2149 | |
| Peptide, recombinant protein | Ubiquitin from bovine erythrocytes | Sigma-aldrich | Sigma-aldrich: U6253 | |
| Peptide, recombinant protein | 1x ubiquitin | this paper | | 1x human ubiquitin (C-terminal His-tagged) |
| Peptide, recombinant protein | 2x ubiquitin | this paper | | 2x tandem linear human ubiquitin (C-terminal His-tagged) |
| Peptide, recombinant protein | 3x ubiquitin | this paper | | 3x tandem linear human ubiquitin (C-terminal His-tagged) |
| Peptide, recombinant protein | 4x ubiquitin | this paper | | 4x tandem linear human ubiquitin (C-terminal His-tagged) |

*Continued on next page*

*Continued*

| Reagent type (species) or resource | Designation | Source or reference | Identifiers | Additional information |
|---|---|---|---|---|
| Peptide, recombinant protein | GST-mRABGEF1 (WT) | this paper | | GST-tagged mouse RABGEF1 (WT) 1-74aa |
| Peptide, recombinant protein | GST-mRABGEF1 (Y26A) | this paper | | GST-tagged mouse RABGEF1 (Y26A) 1-74aa |
| Peptide, recombinant protein | GST-mRABGEF1 (A58D) | this paper | | GST-tagged mouse RABGEF1 (A58D) 1-74aa |
| Peptide, recombinant protein | GST-mRABGEF1 (Y26A/A58D) | this paper | | GST-tagged mouse RABGEF1 (Y26A/A58D) 1-74aa |
| Peptide, recombinant protein | GST-TcPINK1 | *Yamano et al. (2015)* | | |
| Other | Ni-NTA agarose | QIAGEN | QIAGEN: 30230 | |
| Other | PD MidiTrap G-25 | GE Healthcare | GE Healthcare: 28-9180-08 | |
| Other | Glutathione-Sepharose 4B | GE Healthcare | GE Healthcare: 17-0756-01 | |
| Other | Superdex 75 10/300 column | GE Healthcare | GE Healthcare: 17-5174-01 | |
| Other | Amicon Ultra centrifugal filters | Millipore | Millipore: UFC800308 for 3K Millipore: UFC800308 for 10K | |
| Software, algorithm | Photoshop | Adobe | SCR_014199 | |
| Software, algorithm | Volocity | PerkinElmer | SCR_002668 | |
| Software, algorithm | ZEN microscope software | Carl Zeiss | SCR_013672 | |
| Software, algorithm | GraphPad Prism v6.0d | GraphPad Software | SCR_002798 | |

## DNA constructs

Retrovirus plasmids pCHAC/YFP-LC3B-IRES-MCS2, pBMNz/YFP-Parkin, pBMNz/mCherry-Parkin, pBABE-puro/2HA-RAB7A and pBABE-puro/YFP-RAB7A were described previously (*Yamano et al., 2014*). Plasmids for transient expression of HA-TBC1D15, HA-TBC1D17, or YFP-FIS1 were also described previously (*Yamano et al., 2014*). Q67L mutation or C-terminal four amino acids deletion in *RAB7A* gene was introduced by PCR-based mutagenesis using appropriate primers pair and subcloned into pBABE-puro vector (Addgene plasmid 1764). 2 × HA-tagged *RAB7A (T22N)* gene from DsRed-rab7 DN (Addgene plasmid 12662) was subcloned into pBABE-puro vector to generate pBABE-puro/2HA-RAB7A (T22N). *MON1B* and *CCZ1* genes were amplified by PCR from MGC Human MON1B cDNA (MHS6278-202832311 from Dharmacon) and MGC Human CCZ1 cDNA (MHS6278-202808650 from Dharmacon) and subcloned into pEGFP-C1 vector (Clontech, Mountain View, CA) to make pEGFP-MON1B and pEGFP-CCZ1, respectively. When expressing MON1/CCZ1 complex, *EGFP-MON1B* and untagged *CCZ1* genes (or *EGFP-CCZ1* and untagged *MON1B* genes) were subcloned into pCHAC-MCS1-IRES-MCS2 (Allele Biotechnology, San Diego, CA). Y26A and A58D mutations in pEGFP-C1/mouse RABGEF1 plasmid (a kind gift from M. Fukuda) were introduced by PCR-based mutagenesis. *EGFP-mouse RABGEF1 (WT and mutants)* genes were subcloned into pBABE-puro vector and pBMN-Z vector (Addgene plasmid 1734). *Mouse RAB5B, RAB5C, RAB17* and *RAB29* genes (kind gifts from M. Fukuda) were first inserted into BamHI/NotI sites of pBSK(-)/3 × HA (Yamano, unpublished) to put 3 × HA tag sequence at their 5'-regions. *3HA-tagged mRAB5B, mRAB5C, mRAB17*, and *mRAB29* genes were then subcloned into pBABE-puro vector.

Plasmids for bacterial expression of tandem linear ubiquitins were constructed as follows. Human ubiquitin (Ub) gene was amplified by PCR using primer pair, BamHI-Ub-F (5'- GGC Cgg atc cAT GCA GAT TTT CGT GAA AAC −3') and Ub-BglII-XhoI-R (5'- CCG Gct cga gcc cag atc tAC CAC CAC GAA GTC TCA ACA −3'). The amplified DNA fragment was treated with BamHI and XhoI and inserted into BamHI/XhoI sites of pBluscriptII SK(-) vector to make pBSK/1 × Ub. To make

2 × tandem Ub, BamHI/XhoI-treated Ub gene was inserted into BglII/XhoI sites of pBSK/1 × Ub. 3 × Ub and 4 × Ub genes were similarly constructed. The resultant tandem Ub genes were subcloned into BamHI/XhoI sites of pET21a(+) vector (Novagen, Madison, WI) to make pET21a(+)/1 × Ub, pET21a(+)/2 × Ub, pET21a(+)/3 × Ub, and pET21a(+)/4 × Ub. For expression of GST-tagged mouse RABGEF1, mouse RABGEF1 (1-74aa) coding region was subcloned into pGEX-6P-1 vector (GE Healthcare, Piscataway, NJ).

## Antibodies

The following antibodies were used for immunoblotting: rabbit anti-GFP (ab6556; Abcam, Cambridge, MA), mouse anti-MFN2 (ab56889; Abcam), rabbit anti-TOMM20 (sc-11415; Santa Cruz Biotechnology, Dallas, TX), rabbit anti-LC3B (L7543; Sigma, St. Louis, MO), mouse anti-MT-CO2 (ab110258; Abcam), mouse anti-Actin (MAB1501R; Millipore, Bedford, MA), mouse anti-RAB7 (ab50533; Abcam), rabbit anti- RABGEF1 (NBP1-49938; NOVUS BIOLOGICALS, Littleron, CO), mouse anti-CCZ1 (sc-514290; Santa Cruz Biotechnology), mouse anti-ubiquitin (sc-8017; Santa Cruz Biotechnology), and rabbit anti-S65 phosphorylated ubiquitin (described previously [*Koyano et al., 2014*]).

The following antibodies were used for immunostaining: rabbit anti-GFP (A-11122; Invitrogen, Grand Island, NY), mouse anti-GFP (A-11120; Invitrogen), rabbit anti-GFP (ab6556; Abcam), rabbit anti-TOMM20 (sc-11415; Santa Cruz Biotechnology), mouse anti-TOMM20 (sc-17764 Clone F-10; Santa Cruz Biotechnology), mouse anti-HA (M180-3; MBL Life science, Japan), mouse anti-HA (HA.11 Clone 16B12; COVANCE, Berkeley, CA), mouse anti-LAMP2 (sc-18822; Santa Cruz Biotechnology), mouse anti-EEA1 (610457; BD Biosciences, San Jose, CA), mouse anti-GM130 (610822; BD Biosciences), mouse anti-pryruvate dehydrogenase E1-alpha subunit (PDHA1) (ab110334; Abcam), mouse anti-DNA (CBL186; Millipore), rabbit anti-RAB5 (C8B1; Cell Signaling Technology, Beverly, MA), rabbit anti-RAB7 (D95F2; Cell Signaling Technology), purified rabbit anti-TBC1D15 (a kind gift from N. Ishihara), and rabbit anti-ATG9A and anti-ATG16L1 (kind gifts from N. Mizushima).

## RNA interference

siRNA oligos were purchased from QIAGEN (Valencia, CA). The target sequences are as follows: RABGEF1_#7, AACCGGCAAACCAGCATTGAA; MON1A_#2, AGCCGTCAGTGCCATCCATAA; MON1B_#2, CTGGGTGACCTCCAAATTCGA; CCZ1_#11, CCCGGATTTAATGAAGATTCT; RAB7A_#5, CACGTAGGCCTTCAACACAAT. Non-targeting control siRNA were described previously (*Lazarou et al., 2013*). siRNAs were transfected into cells using Lipofectamine RNAiMAX (Invitrogen) according to the manufacture's instruction. After 24 hr of transfection, the medium was changed to fresh medium and the cells were gown for further 24 hr before analysis.

## Cell culture and transfection

In this study, we used HeLa cells and HCT116 cells. HeLa cells were used as a model system owing to their robust expression of endogenous PINK1, and HCT116 cells were chosen because they are largely diploid and amenable to CRISPR/Cas9 KO experiments. Parkin overexpression is required because HeLa cells do not express Parkin and while HCT116 cells express Parkin but not unough to trigger robust mitophagy. HeLa and HEK293T cells were cultured in Dulbecco's Modified Eagle Medium (DMEM) (Life Technologies, Carlsbad, CA and Sigma) supplemented with 10% (v/v) fetal bovine serum, 1 mM sodium pyruvate (Life Technologies), 2 mM glutamine (Life Technologies) or 2 mM GlutaMAX (Life Technologies) and nonessential amino acids (Life Technologies). *FIS1-/-* and *TBC1D15/17* DKO and the corresponding WT HCT116 cells were cultured in McCoy's 5A medium (Life Technologies) supplemented with 10% (v/v) fetal bovine serum, 2 mM glutamine or 2 mM GlutaMAX, and nonessential amino acids. Cells were cultured at 37°C in a 5% $CO_2$ incubator. *FIS1-/-* and *TBC1D15/17* DKO HCT116 cells were engineered previously (*Otera et al., 2010*; *Yamano et al., 2014*). Stable cell lines were established by recombinant retrovirus infection as follows. Vector particles were produced in HEK293T cells grown in a six-well plate by cotransfection with Gag-Pol, VSV-G and the aforementioned retrovirus plasmids. After 12 hr of transfection, the medium was changed to a fresh medium and the cells were further cultivated for 24 hr. The viral

supernatants were then infected into HeLa cells or HCT116 cells with 8 µg/ml polybrene (Sigma-Aldrich). FuGENE6 transfection reagent (Promega, Madison, WI) was used for transient expression.

Valinomycin (Sigma-Aldrich) was used at a final concentration of 10 µM. Oligomycin (Calbiochem) and antimycin A (Sigma) were used at final concentrations of 10 µM and 4 µM, respectively. When cells were treated with valinomycin or Oligomycin/Antimycin A more than 6 hr, 10 µM Q-VD-OPH (SM Biochemicals, Anaheim, CA) was added to block apoptotic cell death. Indole-3-acetic acid (IAA) (Wako, Japan) was used at a final concentration of 500 µM.

## Construction of mAID-knock-in cell lines

HCT116 cell carrying a CMV-OsTIR1 transgene at the safe-harbor AAVS1 locus (HCT116-OsTIR1) were used as a parental strain (*Natsume et al., 2016*). 247 bp of 5' and 247 bp of 3' homology arms for Human RABGEF1 exon10 containing BamHI site in the middle (total 500 bp) was synthesized and subcloned into pUC57-Amp (GENEWIZ, South Plainfield, NJ) to prepare RABGEF1-ex10 donor plasmid. DNA fragments of in-frame mAID cassette harboring neomycin (Neo) or hygromycin (Hygro)-resistant markers were inserted into BamHI site of RABGEF1-ex10 donor plasmid to make RABGEF1-ex10-NeoR donor and RABGEF1-ex10-HygroR donor, respectively. We used PrecisionX Cas9 SmartNuclease System (System Bioscineces, Mountain View, CA) to construct a CRISPR/Cas plasmid. Two DNA oligonucleosides, RX5-CRISPR-Top-F (5'- tgt atg aga cca cGA TCA TCC TGC ATA AAC TTG −3') and RX5-CRISPR-Bottom-R (5'- aaa cCA AGT TTA TGC AGG ATG ATC gtg gtc tca −3') were annealed and introduced into EF1-hspCas9-H1-gRNA linearized SmartNuclease vector according to the manufacture's instruction. The resultant Cas9-gRNA plasmid was transfected into HCT116-OsTIR1 cells with aforementioned RABGEF1-ex10-NeoR donor and RABGEF1-ex10-HygroR donor plasmids using FuGENE HD (Promega). Cells were grown in McCoy's 5A media in the presence of 700 µg/ml G418 (Sigma) and 100 µg/ml Hygromycin B (Invitrogen) until forming colonies. The single colonies were then isolated and total cell lysates were analyzed by immunoblotting with anti-RABGEF1 antibody to select cell lines with mAID tagging at the both alleles.

## Bacterial strains

To construct YH1851, the *serB* deletion *Escherichia coli* strain, the Δ*serB::kan* marker was transferred from JW4351 (*Baba et al., 2006*) to BL21(DE3) (Invitrogen) by P1 transduction. Successful replacement of the chromosomal *serB* gene was confirmed by PCR amplification, kanamycin resistance and amino acid auxotrophy for the mutant strain.

## Immunocytochemistry and confocal imaging

Cells grown on two-well coverglass chamber slides or on MatTek were fixed with 4% paraformaldehyde in PBS for 25 min at room temperature, permeabilized with 0.15%(v/v) TX-100 in PBS for 15 min, and preincubated with 2%(w/v) BSA or 0.1% gelatin in PBS for 30 min. The fixed cells were incubated with primary antibodies and appropriate secondary antibodies (goat anti-rabbit or anti-mouse IgG, Alexa Fluor 488, 568 or 647 conjugated from Invitrogen) for immunostaining. The images of the cells were captured using an inverted confocal microscope (LSM510 Meta, or LSM710, Carl Zeiss, Germany) with a 63×/1.4 NA or 40×/1.4 oil differential interference contrast Plan-Apochromat objective lens. For image analysis, Volocity (PerkinElmer, Waltham, MA), ZEN microscope software (Carl Zeiss) and/or Photoshop (Adobe, San Jose, CA) software were used. To stain nuclear DNA, the cells were incubated with 0.3 µg/ml DAPI (Thermo Fisher Scientific, Waltham, MA) in PBS for 5 min. Volocity software (for *Figure 1G*, *Figure 3E*, *Figure 3F*, and *Figure 4C*) or ZEN software (for *Figure 6D* and *Figure 7E* ) was used for colocalization analysis. Each single cell was first selected as a ROI, and appropriate thresholds of green and ref channels (for example, an HA-RAB7A signal for green channel and a TOMM20 signal for red channels) set in each cell. Overlapped HA-RAB7A signal with a TOMM20 signal per total HA-RAB7A signal intensity in a selected cell (weighted colocalization coeffcients) was measured. ATG9A recruitment to damaged mitochondria was measured as a value of overlapped ATG9A signal with mitochondria-targeted mCherry-Parkin signal. Colocalization of ATG9A with YFP-LC3B was measured as a value of overlapped ATG9A signal with membrane-targeted YFP-LC3B signal. Volocity software was used for the measurement of ATG9A localization ratio between Golgi and cytosol. First, immunostained GM130-positive Golgi region was selected by a circle 8.35 µm in diameter as a ROI, and the signal intensity of immunostained ATG9A overlapped with

the GM130 signal in the ROI were measured for golgi-localized ATG9A. Cytosolic ATG9A signal intensities were calculated from the average of three randomly selected cytosolic areas, each of them is a circle 8.35 µm in diameter. Finally, The ratio between Golgi and cytosol was calculated.

## Immunoblotting

Cells grown in six-well plate were washed twice with PBS and solubilized with 2% CHAPS buffer (25 mM HEPES-KOH pH 7.5, 300 mM NaCl, 2% (w/v) CHAPS, protease inhibitor cocktail (Roche, Indianapolis, IN)) on ice for 30 min and then protein concentrations were determined. Proteins precipitated with TCA were lysed with SDS-PAGE sample buffer supplemented with DTT. The appropriate amounts of proteins were applied and separated on 4–12% Bis-Tris SDS-PAGE (Invitrogen) with MES or MOPS SDS running buffer (Invitrogen). After transfer, PVDF membrane were blocked and incubated with primary antibodies. Proteins were detected using alkaline phosphatase-conjugated goat anti-mouse or anti-rabbit IgG as secondary antibodies and a BCIP-NBT solution kit (Nacalai Tesque, Japan). For detecting phosphorylated ubiquitin, anti-rabbit IgG horseradish peroxidase-linked secondary antibodies (GE Healthcare Life Sciences) and Western Lightning Plus-ECL (PerkinElmer) were used.

## Phos-tag PAGE

For analysis of ubiquitin phosphorylation, Phos-tag (Wako) and $MnCl_2$ were added to polyacrylamide Tris-glycine gels at the levels recommended by the manufacturer.

## Mass spectrometry

*TBC1D15/17* DKO HCT116 cells stably expressing mCherry-Parkin and 2HA-RAB7A (T22N) were treated with DMSO or valinomycin for 3 hr. The harvested cells from four 15 cm dishes (approximately 80% confluent) were then treated with 5 mM DTBP (dimethyl 3,3'-dithiobispropionimidate, Pierce, Rockford, IL) for 10 min at room temperature. The crosslinking reaction was quenched by adding 125 mM glycine. After washing with PBS, the cell pellet was solubilized with MCLB buffer (50 mM Tris pH 7.8, 150 mM NaCl, 0.5% NP40) containing protease inhibitor cocktail (Roche) for 15 min on ice. Cleared cell lysate were then incubated with equilibrated anti-HA beads (Sigma) overnight at 4°C while gently inverting. Anti-HA beads were washed with ice-cold MCLB buffer containing protease inhibitor cocktail once, and with ice-cold PBS buffer three times. The bound proteins were eluted with 250 µg/ml HA peptide (Sigma) in PBS buffer, and then precipitated with 10% trichloroacetic acid. Precipitated proteins were resuspended in 50 mM Tris pH 8 with 10% acetonitrile and disulfide bonds reduced by 5 mM DTT (30 min at 55°C), alkylated with 15 mM iodoacetamide (room temperature, 30 min in the dark), and excess iodoacetamide quenched by the addition of 10 mM DTT (room temperature, 20 min in the dark). Proteins were precipitated using chloroform/methanol precipitation and pellets resuspended in 50 mM Tris pH 8.0 with 10% acetonitrile and digested with 25 ng/µl trypsin overnight at 37°C. Peptides were desalted on C18 stagetips, dried, resuspended in 10 µl 5% acetonitrile, 5% formic acid, and 4 µl (in technical duplicate) loaded onto an 18 cm by 100 µm (inner diameter) C18 column, eluted by a 50 min 8% to 26% acetonitrile gradient and analyzed on a LTQ linear ion trap mass spectrometer (ThermoFinnigan, San Jose, CA) by MS/MS using a top-10 method. Spectra were searched against a target-decoy database of human tryptic peptides including common contaminants using SEQUEST (carbamidomethylation as static modification, oxidation of methionine residues as variable modification) followed by a linear discriminant analysis with a 2% false discovery rate. High-confidence interactors were determined using CompPASS analysis (*Sowa et al., 2009*).

## Recombinant proteins

Recombinant tandem linear ubiquitins were prepared as follows. *Escherichia coli* BL21-CodonPlus (DE3)-RIL competent cells (Agilent Technologies, Santa Clara, CA) transformed with pET21 plasmids encoding 1x, 2x, 3x, and 4x ubiquitins were grown in LB medium supplemented with 100 µg/ml ampicillin and 25 µg/ml chloramphenicol at 37°C. Ubiquitins were expressed at 37°C for 3 hr by addition of 1 mM IPTG. The bacterial cell pellets after centrifugation were resuspended in TBS buffer (50 mM Tris-HCl pH7.5, 120 mM NaCl) supplemented with lysozyme, DNAse I, DTT, $MgCl_2$, and protease inhibitor cocktail and stored at −20°C until used. The frozen cell suspension were thawed and

sonicated (Advanced-Digital Sonifer, Branson), and insoluble proteins were removed by centrifugation. The supernatants were mixed with equilibrated Ni-NTA agarose (QIAGEN) for 30 min at 4°C. The agarose was then loaded onto a column and washed with TBS buffer containing 20 mM imidazole. The bound proteins were eluted with 200 mM imidazole. Imidazole was removed via a PD Midi-Trap G-25 (GE Healthcare).

Recombinant GST-tagged mouse RABGEF1 (1-74aa) (GST-mRABGEF1) WT and mutants were prepared as follows. BL21-CodonPlus(DE3)-RIL cells harboring pGEX-6P-1 plasmids were grown in LB medium supplemented with 100 μg/ml ampicillin and 25 μg/ml chloramphenicol at 37°C. Expression of GST-mRABGEF1 was induced by addition of 50 μM $ZnCl_2$ and 100 μM IPTG for 16 hr at 16°C. The bacterial cell pellets after centrifugation were resuspended in TBS buffer (50 mM Tris-HCl pH7.5, 120 mM NaCl) supplemented with lysozyme, DNAse I, DTT, $MgCl_2$, and protease inhibitor cocktail and stored at −20°C until used. The thawed cell suspension was sonicated, and insoluble proteins were removed by centrifugation. The supernatants were mixed with equilibrated glutathione-Sepharose 4B (GE Healthcare) for 30 min at 4°C. The sepharose was then loaded onto a column and washed with TBS buffer containing 1 mM Tris(2-carboxylethyl)phosphine (TCEP; Sigma). The bound proteins were eluted with TBS buffer containing 1 mM TECP and 20 mM L-glutathione reduced (GSH; Sigma). The GSH was removed and the buffer was replaced with TBS buffer supplemented with 1 mM TCEP and 10%(w/v) glycerol via a PD MidiTrap G-25. Purification of recombinant GST-TcPINK1 is described previously (*Yamano et al., 2015*). In vitro ubiquitin phosphorylation was performed as follows. Mixture of recombinant 1x, 2x, 3x and 4x ubiquitins (each 7.3 μM) was incubated with 2.8 μM recombinant GST-TcPINK1 in kinase buffer (50 mM Tris-HCl pH7.5, 120 mM NaCl, 10 mM ATP, 50 mM $MgCl_2$, 1 mM TCEP) for 2 hr at 32°C.

## K48- and K63-polyubiquitin chains

K48-linked polyubiquitin chains were produced using recombinant E2-25K (*Pickart and Raasi, 2005*). 2.4 mg/ml ubiquitin from bovine erythrocytes (U6253; Sigma) was incubated with recombinant 0.5 μM E1 and 10 μM E2-25K in Buffer (50 mM Tris-HCl pH8.0, 4 mM ATP, 5 mM $MgCl_2$, 1 mM TCEP) at 37°C for 1 hr. When producing K48-linked S65-phosphorylated polyubiquitin chains, the above reaction was done in the presence of 2.85 μM GST-TcPINK1 at 32°C for 2 hr. K63-linked polyubiquitin chains were produced using recombinant Ubc13 and MMS2 (*Sato et al., 2008*). 2.4 mg/ml ubiquitin from bovine erythrocytes (U6253; Sigma) was incubated with recombinant 0.5 μM E1, 5 μM Ubc13 and 5 μM MMS2 in Buffer (50 mM Tris-HCl pH8.0, 4 mM ATP, 5 mM $MgCl_2$, 1 mM TCEP) at 37°C for 5 hr. When producing K63-linked S65-phosphorylated polyubiquitin chains, 2.85 μM GST-TcPINK1 was added to the reaction and further incubated at 32°C for 2 hr.

## In vitro binding assay

Recombinant GST-mRABGEF1 WT and mutants were bound to glutathione sepharose by mixing in TBS buffer containing 1 mM TCEP and 0.1%(v/v) TX-100 for 20 min at 4°C. Unbound GST-mRAB-GEF1 was removed by washing the sepharose three times with TBS buffer containing 1 mM TCEP and 0.1%(v/v) TX-100. Mixture of unphosphorylated or phosphorylated ubiquitins, or K48- or K63-linked unphosphorylated or phosphorylated polyubiquitin chains were then added and incubated for 30 min at 4°C. The resin was washed three times with TBS buffer containing 1 mM TCEP and 0.1%(v/v) TX-100, and proteins were eluted with SDS-PAGE sample buffer.

## Isothermal titration calorimetry

S65 phosphorylated ubiquitin was produced in bacterial cells as follows. To anchor TcPINK1 to the bacterial inner membrane, a DNA fragment encoding 1-40aa of AtpF transmembrane segment was inserted into pACYC-Duet-1/TcPINK1. pET21a(+)/1xUb and pACYC-Duet-1/AtpF-TcPINK1 were introduced into YH1851 cells, and the transformants were grown in LB medium supplemented with 100 μg/ml ampicillin and 25 μg/ml chloramphenicol at 37°C. Expression of phosphorylated ubiquitin was induced by addition of 300 μM IPTG at 32°C for 4 hr. The purification process was essentially the same as that of unphosphorylated ubiquitin except for addition of PhosSTOP phosphatase inhibitor cocktail (Roche) in cell suspension buffer. For ITC measurement, the exact same lot buffer was used. Recombinant ubiquitin, phosphorylated ubiquitin, GST-mRABGEF1 WT and Y26A/A58D were applied onto a Superdex 75 10/300 column (GE Healthcare) equilibrated with TBS buffer, and buffer

exchange was carried out on an AKTA purifier system at a flow rate of 0.4 ml/min. Appropriate protein fractions were collected and concentrated using Amicon Ultra centrifugal filters (Millipore). The binding affinity of RABGEF1 to ubiquitin or phosphorylated ubiquitin was measured using a MicroCal iTC200 calorimeter (GE Healthcare), with stirring at 1000 rpm at 25°C. The titration involved 18 injections of 2 µl of 300 µM ubiquitin or phosphorylated ubiquitin at intervals of 120 s into a sample cell containing 200 µl of 30 µM GST-mRABGEM7 (WT or Y26/A58D). The titration data were analyzed using the MicroCal OriginTM 7.0 software to determine the enthalpy ($\Delta H$), dissociation constant ($K_d$), and stoichiometry of binding (N). Thermal titration data were fit to a single site binding model and thermodynamic parameters $\Delta H$ and $K_d$ were obtained by fitting to the model. The error of each parameter shows the fitting error.

## Immunoelectron microscopy

*TBC1D15/17* DKO HCT116 cells stably expressing YFP-RAB7A and mCherry-Parkin were treated with or without valinomycin for 3 hr and then fixed for 30 min with 4% paraformaldehyde and 0.1% glutaraldehyde in PBS. The fixed cells were washed four times with PBS, followed by permeabilization for 40 min with 0.1% Saponin and 5% goat serum in PBS. The cells were incubated for 1 hr with mouse anti-GFP antibody (A-11120; Invitrogen), followed by 1 hr with nanogold-conjugated anti-mouse IgG antibody (Nanoprobes, Yaphank, NY) and further processing as described previously (*Tanner et al., 1996*). Thin sections (~80 nm) were counter stained with uranyl acetate and lead citrate. The sections were examined with a JEOL 200 CX transmission electron microscope. Images were collected with a digital CCD camera (AMT XR-100; Danvers, MA).

## Mt-mKeima-based mitophagy assay with FACS

YFP-Parkin and mt-mKeima were stably expressed in HCT116-OsTIR1 parental (WT) and *RABGEF1-mAID* cells via retrovirus infection as previously reported (*Lazarou et al., 2015*). Stable cell lines were then seeded in 6-well plates, treated with IAA for 16 hr, then with OAG for 6 hr before harvested for FACS analysis following the procedure as shown in *Lazarou et al. (2015)*.

## Statistical analysis

Error bars represent SE as indicated in the figure legends. Data were processed in Excel and statistical analyses were peformed using Excel or GraphPad Prism v6.0d (GraphPad Software, La Jolla, CA). Statistical analysis of differences between two groups was performed using a two-tailed, unpaired t test (Excel) and between more than two groups using a one-way analysis of Dunnett's multiple comparisons test. N.s., not significant, *p<0.05, ***p<0.001.

## Acknowledgements

We thank Dr. Yusuke Sato for plasmids for bacterial expression of E2-25K, Ubc13 and MMS2, Dr. Yasushi Saeki and Dr. Hikaru Tsuchiya for purification of recombinant E2-25K, Ubc13 and MMS2, Dr. Yukiko Yoshida for statistical analysis, Dr. Noboru Mizushima for anti-ATG9A and anti-ATG16L1 antibodies and retroviral plasmids pMXs-IP-GFP-ULK1, pMXs-IP-EGFP-hAtg13, pMXs-puro GFP-DFCP1 and pMXs-IP GFP-WIPI1, Dr. Mitsunori Fukuda for mouse cDNAs of *RAB5B*, *RAB5C*, *RAB17*, *RAB29*, and *RABGEF1*, Dr. Jean-François Trempe for GST-TcPINK1 plasmid, Dr. Naotada Ishihara for purified anti-TBC1D15 antibodies, Dr. Dragan Maric for cell sorting, and members of Youle laboratory and ubiquitin project for valuable discussions and comments.

---

## Additional information

### Competing interests

Richard J Youle: Reviewing editor, *eLife*. Wade Harper: Reviewing editor, *eLife*. The other authors declare that no competing interests exist.

## Funding

| Funder | Grant reference number | Author |
| --- | --- | --- |
| Japan Society for the Promotion of Science | JP16K18545 | Koji Yamano |
| Japan Science and Technology Agency | JPMJCR13M7 | Nobuo N Noda |
| Takeda Science Foundation | | Keiji Tanaka<br>Noriyuki Matsuda |
| Chieko Iwanaga Fund for Parkinson's Disease Research | | Noriyuki Matsuda |
| National Institute of Neurological Disorders and Stroke | Intramural program | Koji Yamano<br>Chunxin Wang<br>Shireen A Sarraf<br>Richard J Youle |
| Japan Society for the Promotion of Science | JP26111729 | Noriyuki Matsuda |
| Japan Society for the Promotion of Science | JP15H01196 | Noriyuki Matsuda |
| Japan Society for the Promotion of Science | JP26000014 | Keiji Tanaka |
| Japan Society for the Promotion of Science | JP26840033 | Yohei Hizukuri |
| Japan Society for the Promotion of Science | 16K15095 | Masato T. Kanemaki |

The authors declare that there was no funding for this work.

## Author contributions

Koji Yamano, Conceptualization, Formal analysis, Funding acquisition, Investigation, Visualization, Writing—original draft, Project administration, Writing—review and editing; Chunxin Wang, Shireen A Sarraf, Christian Münch, Formal analysis, Investigation, Writing—review and editing; Reika Kikuchi, Investigation, Writing—review and editing; Nobuo N Noda, Yohei Hizukuri, Resources, Investigation, Writing—review and editing; Masato T Kanemaki, Resources, Methodology, Writing—review and editing; Wade Harper, Supervision, Methodology, Writing—review and editing; Keiji Tanaka, Supervision, Funding acquisition, Writing—original draft, Writing—review and editing; Noriyuki Matsuda, Funding acquisition, Writing—original draft, Project administration, Writing—review and editing; Richard J Youle, Conceptualization, Supervision, Funding acquisition, Writing—original draft, Writing—review and editing

## Author ORCIDs

Koji Yamano http://orcid.org/0000-0002-4692-161X
Christian Münch http://orcid.org/0000-0003-3832-090X
Nobuo N Noda https://orcid.org/0000-0002-6940-8069
Wade Harper http://orcid.org/0000-0002-6944-7236
Richard J Youle http://orcid.org/0000-0001-9117-5241

## Decision letter and Author response

Decision letter https://doi.org/10.7554/eLife.31326.042
Author response https://doi.org/10.7554/eLife.31326.043

## Additional files

### Supplementary files

• Supplementary file 1. Proteomic analysis of 2HA-RAB7A (T22N)-associated proteins during mitophagy. This files contains all raw and analyzed mass spectrometric data and analysis parameters. Proteomic analysis of 2HA-RAB7A (T22N)-associated proteins in *TBC1D15/17* DKO HCT116 cells stably

expressing mCherry-Parkin after 3 hr of valinomycin treatment using CompPASS. The tab labeled 'Analysis' contains information regarding cell lines used, experimental conditions, descriptions of all worksheets including raw data that contain the complete lists of all proteins identified, WDN-scores, Z-scores, and APSMs, and details of each subsequent analysis performed.

DOI: https://doi.org/10.7554/eLife.31326.039

• Transparent reporting form
DOI: https://doi.org/10.7554/eLife.31326.040

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
