## [Decision Letter]

Thank you for submitting your article "Endosomal Rab cycles regulate Parkin-mediated mitophagy through ATG9A assembly" for consideration by *eLife*. Your article has been reviewed by two peer reviewers, and the evaluation has been overseen by Ivan Dikic as the Senior and Reviewing Editor. The reviewers have opted to remain anonymous.

The reviewers have discussed the reviews with one another and the Reviewing Editor has drafted this decision to help you prepare a revised submission.

SUMMARY:

Yamano et al. examined the role of Rab GTPases in PINK1/Parkin-dependent mitophagy. Building on previous work on the two mitochondrial Rab GAPs TBC1D15 and TBC1D17, the authors showed that loss of TBC1D17/TBC1D15 or their mitochondrial receptor FIS1 leads to a partial translocation of Rab7 from lysosomes to the outer mitochondrial membrane (OMM) in a manner dependent on Rab7's GTPase activity and prenylation acceptor site as well as on Parkin overexpression in response to valinomycin treatment. Given the presence of Rab7 on mitochondria under these conditions, the authors probed whether Rab7 is required for mitophagy. Indeed, the authors observed a reduction in spherical LC3 structures containing the mitochondrial matrix protein PDH as well as a blocked turnover of TOMM20, PDH and mitochondrial DNA. Moreover, ATG9 failed to translocate to mitochondria upon valinomycin treatment in Rab7 knockdown cells. Conversely, the authors scored a pronounced colocalization of ATG9 and LC3B in cells lacking TBC1D15/TBC1D17, which feature increased levels of LC3B nearby damaged mitochondria. By performing interaction proteomics, the authors identified the MON1/CCZ1 complex as mitophagy-dependent Rab7 binding partner, which mediate mitochondrial translocation of Rab7. Focusing on the upstream Rab GTPase of MON1/CCZ1, the authors showed that Rab5 is likewise recruited to damaged mitochondria. Intriguingly, the authors went on to demonstrate that mitochondrial Rab5 translocation is dependent on the ubiquitin-binding domain of RABEX5, which in turn is recruited to mitochondria by Parkin-induced ubiquitylation of OMM proteins. Finally, the authors found that RABEX5 contributes to Oligomycin/AntimycinA-induced mitophagy. Together, this work establishes the involvement of a ubiquitin-directed Rab cascade in the autophagic engulfment of damaged mitochondria. This is an unexpected finding which has implication for other types of selective autophagy.

Essential revisions:

1) Most importantly, the authors do not provide any evidence that endogenous Rab7 and/or endogenous Rab5 is recruited to damaged mitochondria. Such experiments would significantly alter the impact of the authors' findings.

2) To finally prove the cascade of events proposed, please show that Rabex-5 and Rab5 are still recruited to OMM of damaged mitochondria upon ablation of Rab7 or MON1/CCZ1. This will confirm that Rab7 and its GEFs are downstream Rabex/Rab5.

3) Where in the Rab cascade do the authors place TBC1D15 and TBC1D17? Does depletion of Rab5, Rab7, RABEX5 or MON1A/B&CCZ1 effect the mitochondrial localization of these two Rab GAPs?

---

## [Author Response]

Essential revisions:1) Most importantly, the authors do not provide any evidence that endogenous Rab7 and/or endogenous Rab5 is recruited to damaged mitochondria. Such experiments would significantly alter the impact of the authors' findings.

We agree that it would be important to observe mitochondrial recruitment of endogenous RAB5 and RAB7. For this purpose, we obtained commercially available anti-RAB5 and anti-RAB7 antibodies. First, we tested whether these antibodies can be used for immunostaining, and found that the signal stained with anti-RAB5 antibody overlapped with the early endosome marker EEA1 (Figure 7 in the revised manuscript), and the signal stained with anti-RAB7 antibody overlapped with the late endosome/lysosome marker LAMP2 (Figure 1—figure supplement 2 in the revised manuscript). We tested the Rab7 antibody by knocking down endogenous RAB7A by siRNA and showing that the immunostain signal using anti-RAB7 antibody was reduced (Figure 1—figure supplement 2 in the revised manuscript). These results assure us that anti-RAB5 and anti-RAB7 antibodies we used in the revised study properly recognize endogenous RAB5 and RAB7A, respectively.

To test whether endogenous RAB5 and RAB7 are recruited to damaged mitochondria, we used WT, *FIS1-/-*, and *TBC1D15/17* DKO HCT116 cells stably expressing mCherry-Parkin. Under normal growing conditions, neither RAB5 nor RAB7A signals were overlapped with mitochondria (Figure 7 for RAB5 and Figure 1—figure supplement 2 for RAB7A). On the other hand, after mitophagy stimulation by 3hrs of valinomycin treatment, both endogenous RAB5 and RAB7A were recruited to damaged mitochondria especially in *FIS1-/-* and *TBC1D15/17* DKO cells (Figure 7 for RAB5 and Figure 1—figure supplement 2 for RAB7A). These results indicate that not only exogenous, but also endogenous RAB5 and RAB7A are recruited to damaged mitochondria during mitophagy, and loss of mitochondrial Rab-GAPs (TBC1D15 and TBC1D17) stabilizes the association.

2) To finally prove the cascade of events proposed, please show that Rabex-5 and Rab5 are still recruited to OMM of damaged mitochondria upon ablation of Rab7 or MON1/CCZ1. This will confirm that Rab7 and its GEFs are downstream Rabex/Rab5.

In the original manuscript, we have shown that RAB5C recruitment to mitochondria was not inhibited by knocking down of MON1/CCZ1 complex (original Figure 7). Now, we show that knocking down RAB7A did not affect the RAB5 recruitment to mitochondria as well (Figure 7 in the revised manuscript). Furthermore, we tested whether RABGEF1 (Rabex5) recruitment was inhibited by downstream Rabs and Rab-related proteins. As shown in Figure 8—figure supplement 2 in the revised manuscript, in both HeLa and HCT116 cells, neither knocking down MON1/CCZ1 complex nor RAB7A abrogated RABGEF1 recruitment. Therefore, in addition to the original data, these results indicate that RABGEF1 recruitment to damaged mitochondria depends on poly-ubiquitin chains on the surface of damaged mitochondria, but not the downstream Rabs or Rab-related factors.

3) Where in the Rab cascade do the authors place TBC1D15 and TBC1D17? Does depletion of Rab5, Rab7, RABEX5 or MON1A/B&CCZ1 effect the mitochondrial localization of these two Rab GAPs?

TBC1D15 and TBC1D17 were identified as FIS1-binding proteins (Onoue et al., JCS 2013 and Yamano et al., *eLife* 2014). Indeed, although overexpressed TBC1D15 and TBC1D17 localize in the cytosol, overexpressed FIS1 relocates them to mitochondria (Figure 10 in the revised manuscript). Therefore, mitochondrial localization of these two Rab-GAPs seems to depend on FIS1. However, in order to exclude the possibility that endosomal Rab cycles contribute to the localization of TBC1D15, we knocked down the upstream GEFs and Rabs (RABGEF1, MON1, CCZ1, and RAB7A) and observed endogenous TBC1D15 during mitophagy by immunostaining. As shown in the Figure 10 in the revised manuscript, TBC1D15 still localizes on the mitochondria in cells treated with endosomal Rab siRNAs.